# Characteristics of wild hazelnut populations in Northeast China and selection of superior provenances

**Qiwen Yuan**[ID]**, Yang Chen, Dongyang Zhang, Siyu Yang, Minghui Yang, Xuesong Zhu, Chunyu Guan**[ID]*

College of Life Sciences, Agriculture and Forestry, Qiqihar University, Qiqihar, China

* gcy19810201@163.com

**Data Availability Statement:** All relevant data are within the manuscript and its Supporting Information files.

## Abstract

China possesses a copious and geographically diverse reservoir of hazel (*Corylus* spp.) resources, albeit scholarly endeavors focusing on the selection and domestication of indigenous wild hazelnut strains remain scant. To develop and utilize high-quality wild hazelnut resources, this study selected eight populations of wild hazelnuts from seven different provenances in Heilongjiang Province, China. Natural hybrid seeds of eight populations were sown in the Chohai Forest Farm in Longjiang County, Heilongjiang Province, in 2018. In April 2020, two-year-old seedlings were used to establish a provenance trial forest, thereby initiating the provenance trial. Growth parameters were measured using electronic calipers, and pollen characteristics were observed under an electron microscope. The trials meticulously explored the trees' flowering biology, growth parameters, and the economic attributes of the produced nuts. Principal component analysis was employed to comprehensively assess differences among the wild hazelnut populations from various provenances, aiming to identify superior wild hazelnut provenances. The results unveiled significant geographical variations among the tested provenances across several facets: the flowering period of male blossoms extended from April 2nd to April 19th, while female flowering occurred within the timeframe of April 2nd to April 22nd. Moreover, pollen grain morphology demonstrated variability, with the polar axis ranging from 14.02 to 17.09 micrometers, the equatorial axis spanning 16.02 to 18.75 micrometers, and the ratio of polar to equatorial axes fluctuating between 0.88 and 0.92. Through correlation analyses, nut length emerged as a pivotal determinant significantly influencing both kernel weight and the hundred-grain weight. A principal component analysis (PCA) further consolidated these findings, selecting the Nehe-originated variegated *Corylus heterophylla* as the superior provenance based on a comprehensive evaluation of its combined traits. This study constitutes a seminal contribution to the hazelnut breeding endeavor in China, establishing a robust foundation for informed cultivation strategies geared towards optimizing both the yield and the quality of hazelnut resources, thereby advancing the understanding and exploitation of China's wild hazelnut biodiversity.

**Funding:** This research was funded by the Basic Research Fees of Universities in Heilongjiang Province, China (145309628), Graduate Innovation Project of Qiqihar University (X202310232032), Graduate Innovation Project of Qiqihar University (X202310232040), Heilongjiang Provincial Natural Science Foundation of China (LH2022C102). All funders provided financial support.

**Competing interests:** The authors have declared that no competing interests exist.

## 1. Introduction

*Corylus heterophylla* Fisch., a member of the Betulaceae family and classified within the genus Hazel, holds a distinguished position among the "Four Great Nuts of the World." Characterized as a diploid organism with a chromosome count of 2n = 22 [1], this species exists as a perennial, deciduous shrub or small tree exhibiting monoecious, wind-pollinated, unisexual reproductive features [2, 3]. In China, *Corylus heterophylla* Fisch. assumes great ecological and economic significance [4], where its timber, recognized for its hardness, attractive grain, and hue, serves versatile roles in construction, furniture crafting, and minor carpentry applications [5]. Hazelnuts derived from this species are nutritionally dense, replete with unsaturated fatty acids, high-quality proteins, vitamins, minerals, and a variety of amino acids [6–8]. Their versatility extends to a myriad of food products, including hazelnut paste, chopped nuts, flour, chocolates, confectionaries, ice cream, and various bakery items. Notably, consistent consumption of hazelnuts has been associated with a reduced risk of Alzheimer's disease [9]. Beyond their culinary uses, hazelnuts offer additional industrial and medicinal benefits. The shells are repurposed for the manufacture of activated carbon, while the presence of paclitaxel in the kernels holds promise in oncological treatments [10–13]. The species' robust root system facilitates its recognition as an exceptional candidate for soil and water conservation efforts, as well as for improving woodland soils, leading to its worldwide cultivation [5]. Its roots form a symbiosis with truffles, making the species applicable for hazelnut and truffle production. Beyond these uses, studies have shown that Corylus species have multifaceted applications; utilizing hazel crop by-products (i.e., leaves, skins, shells, husks and pruning material) can promote a circular economy and foster sustainable development [14–16].

The primary objectives in the cultivation and breeding of nut trees are to achieve high fruit quality, high yield, and strong disease resistance, thereby creating greater economic value [17–19]. To date, the majority of global research on hazelnut cultivation has concentrated on cultivation techniques, whereas studies focusing on hazelnut rootstocks remain limited. A study conducted in Spain highlighted the advantages of new hazelnut rootstock varieties over self-rooted cultivars [20]. Utilizing robust rootstocks that do not produce suckers is crucial for improving the economic viability of hazelnut cultivation [21, 22]. Germplasm evaluation, hybridization and omics studies are the main breeding strategies [23]. Within the realm of forest tree genetics and breeding, alongside advancements in genome sequencing and transcriptomics, several molecular marker technologies have been developed to support genetic breeding programs. These technologies have enabled research into areas such as genetic diversity and population structure [24–28]. The application of these molecular markers, particularly Simple Sequence Repeat (SSR) markers in hazelnuts, facilitates the creation of DNA fingerprint databases. Such databases are instrumental in authenticating unidentified hazel species, collecting unique germplasm resources, reducing duration and costs of a breeding program and advancing future tree improvement initiatives [4, 29–32].

Due to strong economic returns and the nutritional value of their products, the production and consumption of tree nuts have seen a sharp increase [21]. Presently, the global hazelnut industry witnesses a sustained phase of production and trade expansion. However, the current growth rate of hazelnut production struggles to meet the escalating consumer demand. China stands as the world's foremost importer of shelled hazelnuts, having accounted for roughly half of the global import volume in 2019, thereby highlighting ample room for further exploration and development within its domestic hazelnut industry [33]. Despite being endowed with abundant hazelnut resources, a significant proportion of China's hazelnut trees exist in a wild state, characterized by low yields and a high prevalence of pests and diseases. Northeast China lacks artificially managed native hazelnut forests, with yields from these unmanaged lands

averaging a meager 10 to 20 kilograms per 667 square meters, and in some cases, falling beneath the 10-kilogram threshold. Pest and disease issues exacerbate this situation, with infestation rates soaring to as high as 25%-50%, constituting a pivotal challenge to both the quality and yield of wild-grown hazelnuts [34]. There are two species of hazelnut in Heilongjiang province, Northeast China, namely *Corylus heterophylla* and *Corylus mandshurica*, which are distributed across 24 counties [35]. The area covered by wild hazelnut forests amounts to 782,700 hectares, positioning Heilongjiang as the foremost region in China in terms of both hazelnut resource endowment and productivity.

Attaching importance to the excavation and utilization of wild hazelnut re-sources is an effective way to promote hazelnut breeding. The inter-specific hybridization potential within the genus Corylus is substantial, with wild species offering key traits essential for the development of improved, broadly adaptive cultivars to satisfy the escalating demand for high-quality nut production and diversified uses in ornamental horticulture and other applications [36]. China's hazelnut industry had a belated commencement, only truly transcending its history of non-cultivated hazelnuts with the successful cultivation of superior Eurasian hybrid hazelnut (*Corylus heterophylla* × *C. avellana*) varieties during the 1980s and 1990s. Currently, cultivated hazelnuts in China predominantly consist of *Corylus heterophylla* × *Corylus avellana* hybrids, primarily inhabiting regions north of the Huai River. Conversely, the development of suitably adapted cultivars south of the Qinling Mountains and in the middle-lower reaches of the Yangtze River remains an uncharted territory. Apart from the documented "Dragon Hazel No.1," "Dragon Hazel No.2," and "Dragon Hazel No.3," there is a paucity of literature detailing the selection and breeding of hazel species in Northeast China's Heilongjiang Province [37–39].

Wild hazelnuts in Northeast China, as a valuable forest resource, harbor rich genetic diversity. Due to climate change and variations in plant growth environments, such as low temperatures and drought, different provenances exhibit significant geographic variations and unique biological characteristics. This variability not only shapes the ecological adaptability of hazelnuts but also endows them with distinct advantages in terms of flavor, nutritional value, and appearance, thereby directly influencing their market competitiveness and economic value [17, 40–43]. Cultivars selected from local species reflect the genetic relationships within the germplasm and may indirectly characterize its diversity. Study of genetic diversity is the important stage of germplasm management for increasing the breeding efficacy [44]. This study embarks on a thorough investigation of eight distinct wild hazelnut populations sourced from seven different provenances in Heilongjiang Province, Northeast China. By scrutinizing their flowering biology, growth parameters, and economically relevant nut characteristics, coupled with the implementation of principal component analysis for superior provenance selection. This research lays the foundation for the genetic improvement of hazelnut tree species and the promotion of industrial development.

## 2. Materials and methods

### 2.1. Provenance trial forests

The pollen material, leaf material, and nuts used in this study were obtained from hazelnut provenance trial forests. Florescence observations and growth surveys were also conducted in this place.

**2.1.1 Source of material.** Seeds of wild hazelnut resources in Northeast China were collected in 2018, containing *Corylus mandshurica* and *Corylus heterophylla* of different provenance. For this experiment, seed trees were selected based on criteria that included the absence of severe pests and diseases and higher productivity. Eight hazelnut populations with some excellent traits distributed in seven different geographic provenance sources within the

**Table 1. Participating wild hazelnut populations number information.**

| Abbreviation | Populations | Species |
|---|---|---|
| NJ | Heihe Nenjiang | *Corylus mandshurica* |
| TL | Yichun Tieli | *Corylus heterophylla* |
| NH1 | Qiqihar Nehe | *Corylus heterophylla* |
| NH2 | Qiqihar Nehe | *Corylus heterophylla* |
| AH | Heihe Aihui | *Corylus heterophylla* |
| LJ | Qiqihar Longjiang | *Corylus heterophylla* |
| MDJ | Mudanjiang | *Corylus heterophylla* |
| XK | Heihe Xunke | *Corylus heterophylla* |

Heilongjiang Province were selected, as shown in Table 1. The eight populations of the two species come from seven provenances. *Corylus mandshurica*: One population from Nenjiang. *Corylus heterophylla*: Seven different populations from Tieli, Nehe (with a variant having a striped fruiting testa), Aihui, Longjiang, Mudanjiang, and Xunke. Thirty dominant trees were selected from each geographical source to serve as seed trees, and their natural hybrid seeds were collected. These seeds were sown in the fall of 2018. Seedlings were nurtured in the Cuohai Forest Farm in Longjiang County, Heilongjiang province (E122˚24'55"~122˚56'50", N47˚30'03" ~ 47˚14'55"), to create provenance trial forests in April 2020 using 2-year-old seedlings. The provenance trial forests were 0.67hm$^2$ and a randomized complete block design was used to design four blocks with three replications, in each replication the number of rows of each provenance varied from 1–12 rows with 10 plants per row and a spacing of 1.5m x 3m.

**2.1.2 Overview of the experimental site.** The location of the provenance test forests is Longjiang County Cuohai Forest Farm located in the western part of Heilongjiang Province, Northeast China (E122˚51′, N 47˚27′). The altitude of experimental area is 340 meters. The region experiences a continental monsoon climate, characterized by cold, dry winters influenced by the Siberian High Pressure system, which is accompanied by prevalent northwesterly winds. Summers are notably brief, hot, and rainy. Springs are windy and parched, with meager precipitation, while autumns rapidly cool, witnessing an increased frequency of early frosts. The average annual temperature is 3.4˚C, ≥10˚C throughout the year, the cumulative temperature is 2450˚C-2600˚C.The frost-free period is about 125 days, and the annual precipitation is about 420-480mm.

## 2.2. Studies in flowering biology

**2.2.1 Florescence observation.** Recording of phenological phase by means of point observations method. Flowering observations of hazelnuts from various provenances were conducted in the early spring of 2023 at the experimental site. For each provenance, three robust, undamaged plants exhibiting healthy growth were selected as representatives. Daily observations were systematically conducted between 9:00 and 10:00 hours. To standardize the observational data, dates were transformed into a sequential numerical format, set 2023-04-02 as 1, 2023-04-03 as 2, and so on 2023-04-04 as 3,.. .. ..2023-04-22 as 21, and applied the mathematical and statistical formula for the calculation of the coefficient of variation, the formula was as follows (1):

$$V = \frac{S}{\overline{X}}, \tag{1}$$

In the Formula (1): "V" denotes the coefficient of variation, "S" denotes the standard deviation, and "$\bar{X}$" denotes the average of the numbers in each column.

**2.2.2 Pollen submicroscopic structure.** Pollen grains harvested from hazelnut trees of various provenances were assembled for an in-depth investigation of their submicroscopic morphologies, employing the technology of scanning electron microscopy (SEM). The preparation of SEM specimens entailed the following protocol: The pollen samples underwent a cleansing process utilizing 0.9% saline solution, followed by fixation in a 2.5% glutaraldehyde solution adjusted to a pH range of 7.2 to 7.4. Dehydration of the samples was meticulously carried out via a graded ethanol series. Subsequently, vacuum thermo-stabilized drying methodology was employed to ensure thorough desiccation of the specimens. Thereafter, an aliquot of pollen, approximately 5 mm in slide edge length, was evenly dispersed and coated. Gold sputtering was applied to enhance sample conductivity and visualization. The submicron structural characteristics of the pollen were meticulously examined utilizing a scanning electron microscope (model Apreo S HiVac, courtesy of Thermo Fisher Scientific), with imaging parameters set at an accelerating voltage of 5 kV and a beam current of 0.1 nA to optimize resolution and clarity.

## 2.3. Growth parameters

The current year's growth of the eight participating hazelnut populations was observed after the cessation of plant growth in the current year. The tree height was measured using a standard meter. The ground diameter was measured using electronic vernier calipers. Observations pertaining to overwintering were conducted with the objective of identifying cold-resistant populations. Overwintering success was assessed by the emergence of leaf flush on the extant branches from the previous growth season.

## 2.4. Nut economic traits

In the autumn of 2023, nuts from each hazelnut populations were harvested, and a random subset of 30 nuts per populations was selected for morphometric analysis. Nut dimensions, comprising width, length, thickness, and shell thickness, were meticulously measured utilizing electronic vernier calipers to ensure precision. Furthermore, the nut weight, kernel weight, and hundred-grain weight were accurately quantified using high-precision analytical scales. The kernel percentage was calculated with the following Formula (2):

$$\text{Kernel percentage} = \frac{\text{Kernel weight(g)}}{\text{Nut weight(g)}} \times 100\%. \tag{2}$$

## 2.5. Tree populations selection

As a commercially significant tree populations, hazelnut's nut economic characteristics serve as paramount criteria for assessing tree excellence. These characteristics are multifaceted, with numerous factors contributing to their overall superiority; thus, reliance on a solitary economic trait parameter is insufficient for determining the level of excellence. Principal Component Analysis (PCA) was employed in this study to consolidate the numerous initial indicators into a compact set of representative, composite indices, thereby facilitating a comprehensive evaluation of every tree populations' performance. The data were standardized prior to performing principal component analysis. The experimental data were analyzed using Microsoft

Excel and IBM SPSS Statistics 19.0 to determine means, standard deviations, and significance levels. Graphs and charts were created using Microsoft Excel and Origin 2024.

## 2.6. Populations characterization

Following the rigorous tree selection process, wild hazelnut populations exhibiting exceptional traits were designated as target populations for further examination. Further observed their leaves, branches and other morphological features and determined the nutrient content of the kernels (this work was entrusted to Qingdao Standard Testing Group Co., Ltd.).

## 3. Results

### 3.1. Studies in flowering biology

**3.1.1 Male florescence.** The male hazelnut flowers are catkins. Table 2 presents the morphological characteristics of male inflorescences across eight hazelnut populations. Analysis revealed that NH1 had the longest male inflorescence length both before and after opening. Before opening, TL had the widest male inflorescence at 3.5 mm, while after opening, LJ had the widest inflorescence at 3.6 mm. TL showed the greatest increase in male inflorescence length. TL and LJ had the highest male inflorescence opening rates at 95%, while AH had the lowest at 40%. NH2 had the highest opening rate at 97%, and TL and NH1 had the lowest at 66%. Pollen quantities were low across all populations. Differences were observed in the length, width, flowering rate, and openness of male flower sequences among hazelnut varieties. Notably, varieties TL and LJ exhibited prominence across multiple indices. Measurements of male florescence have been included in the supporting information for this study, please refer to "S1 Data".

Table 3 displays the results of the male inflorescence observations for the participating populations, indicating variability in flowering phenology among the eight hazelnut populations. The initial flowering stage for all populations began on April 2, with no significant differences noted. NH1 and NH2 exhibited the shortest flowering period of 12 days, whereas LJ displayed the longest period of 18 days. NH2 exhibited the longest duration of blooming and the highest flowering intensity. The coefficient of variation for the initial flowering stage was 0, indicating uniform timing of male flower emergence. The flowering period had the highest coefficient of variation at 0.15, suggesting differences in flowering duration among the populations. Combined with Fig 1, these results suggest that the flowering phenology of the male inflorescences across different populations was similar.

**Table 2. Comparison of morphological characteristics of male inflorescences of hazelnut populations.**

| Populations | Before adorable | | blossom/% | After elongation | | Degree of openness/% | Length increased multiple |
|---|---|---|---|---|---|---|---|
| | length(cm) | Width(mm) | | length(cm) | Width(mm) | | |
| NJ | — | — | — | — | — | — | — |
| TL | 1.1±0.25 | 3.5±0.2 | 95 | 2.2±0.4 | 3.1±0.2 | 66 | 2.0 |
| NH1 | 1.8±0.3 | 3.3±0.3 | 85 | 3.2±0.5 | 3.4±0.4 | 66 | 1.8 |
| NH2 | 1.8±0.4 | 3.2±0.2 | 90 | 3.1±0.4 | 3.5±0.2 | 97 | 1.7 |
| AH | 1.6±0.2 | 3.4±0.2 | 40 | 3.0±0.2 | 2.8±0.2 | 88 | 1.9 |
| LJ | 1.7±0.4 | 3.3±0.3 | 95 | 2.9±0.4 | 3.6±0.5 | 78 | 1.7 |
| MDJ | 1.7±0.2 | 3.3±0.3 | 50 | 2.9±0.8 | 3.5±0.2 | 94 | 1.7 |
| XK | 1.6±0.5 | 3.3±0.1 | 90 | 2.4±0.3 | 3.2±0.3 | 88 | 1.5 |

Note: "-" indicates no bloom.

**Table 3. Comparison of male florescence among hazelnut populations.**

| Populations | Male flowers(month/day) | | | Flowering period (day) | Flowering intensity |
|---|---|---|---|---|---|
| | Initial flowering stage | Blooming stage | End stage | | |
| NJ | - | - | - | - | - |
| TL | 4/2 | 4/28 | 4/16 | 15 | Sparse flowering |
| NH1 | 4/2 | 4/7 | 4/13 | 12 | Sparse flowering |
| NH2 | 4/2 | 4/7 | 4/13 | 12 | Profuse flowering |
| AH | 4/2 | 4/10 | 4/17 | 16 | Sparse flowering |
| LJ | 4/2 | 4/8 | 4/19 | 18 | Sparse flowering |
| MDJ | 4/2 | 4/7 | 4/17 | 16flo | Profuse flowering |
| XK | 4/2 | 4/8 | 4/18 | 17 | Moderate flowering |
| Extreme deviation | 0 | 3 | 6 | 6 | |
| Coefficient of variation | 0 | 0.14 | 0.14 | 0.15 | |

Note: "-" indicates no bloom.

**3.1.2 Female florescence.** Table 4 presents a comparison of female flowering stage observations among eight hazelnut populations. MDJ and XK exhibited the shortest florescence of 12 days, while NH2 had the longest florescence at 19 days. The coefficient of variation of

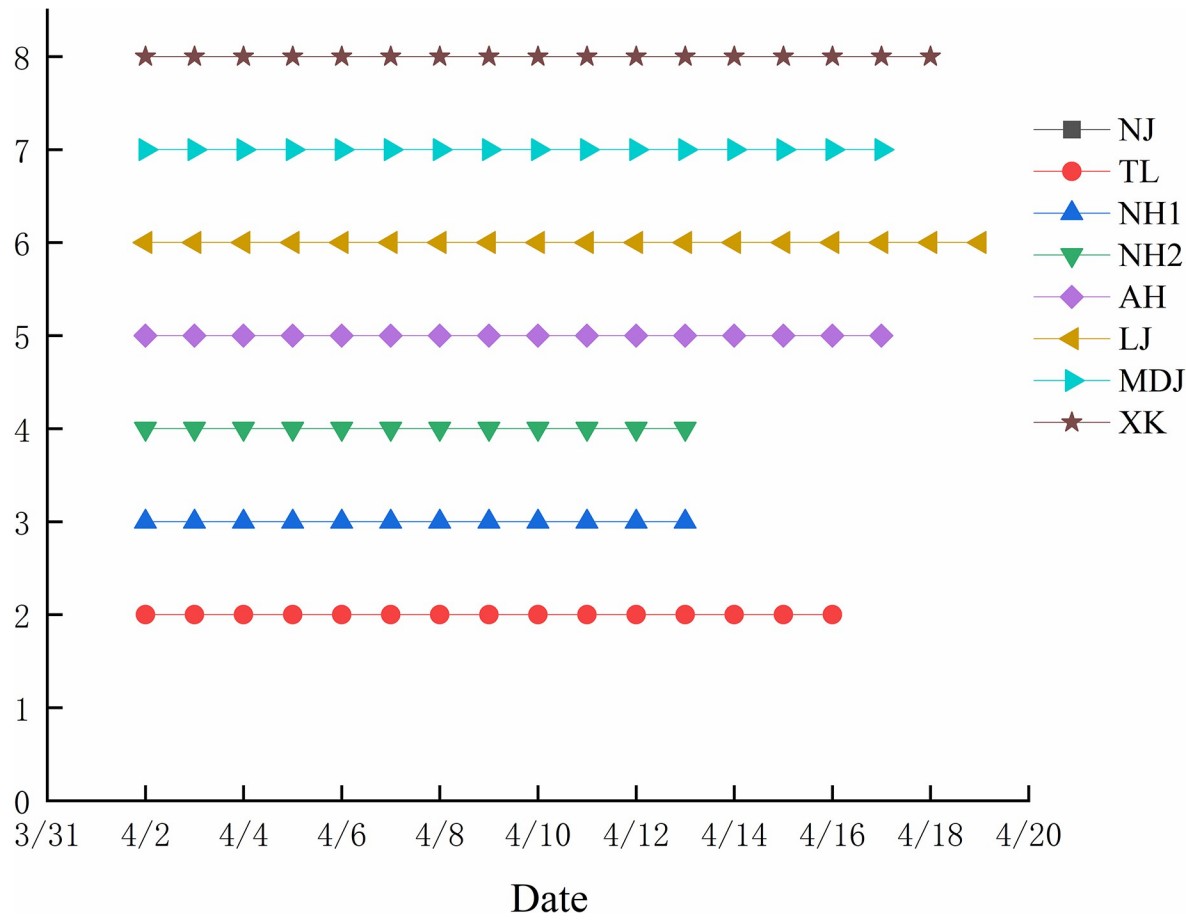

**Fig 1. Comparison of male inflorescence of different populations.** Note: NJ is not flowering.

**Table 4. Comparison of female florescence of nine hazelnut populations.**

| Populations | Female flowers(month/day) | | | Flowering period (day) | Flowering intensity |
|---|---|---|---|---|---|
| | Initial flowering stage | Blooming stage | End stage | | |
| NJ | - | - | - | - | - |
| TL | 4/7 | 4/13 | 4/22 | 16 | Sparse flowering |
| NH1 | 4/9 | 4/13 | 4/21 | 13 | Sparse flowering |
| NH2 | 4/2 | 4/9 | 4/20 | 19 | Sparse flowering |
| AH | 4/10 | 4/15 | 4/22 | 13 | Sparse flowering |
| LJ | 4/9 | 4/13 | 4/22 | 14 | Sparse flowering |
| MDJ | 4/2 | 4/9 | 4/13 | 12 | Sparse flowering |
| XK | 4/11 | 4/16 | 4/22 | 12 | Sparse flowering |
| Extreme deviation | 9 | 7 | 9 | 7 | |
| Coefficient of variation | 0.56 | 0.22 | 0.16 | 0.18 | |

Note: "-" indicates no bloom.

female flowers in the initial flowering stage was the largest at 0.56, and the coefficient of variation in the end stage was the smallest at 0.16, indicating that the female flowers of each hazelnut populations had the most dispersed time distribution in the initial flowering stage, and the most concentrated time distribution in the end stage. From the data combined with Fig 2, it can be concluded that the female flowers of each specie had different of bloom and fall times, which contrasted with the similarity of the male florescence.

**3.1.3 Pollen submicroscopic structure.** Pollen morphology of eight hazelnut populations was examined using scanning electron microscopy, as shown in Fig 3. All pollen grains were single and exhibited similar morphology, being spherical or nearly spherical in both polar and equatorial views. The surface featured uniformly distributed granular bulges and scattered larger nodular granules. Each grain possessed three adjacent germination apertures. Table 5 presents the pollen measurements, indicating polar axis lengths ranging from 14.02 to 17.09 μm and equatorial axis lengths from 16.02 to 18.75 μm. Significant differences ($P < 0.05$) were observed in both polar and equatorial axis lengths, indicating geographic variation in pollen size among populations, with NJ showing the most pronounced differences. Specific measurements of hazelnut pollen polar axis and equatorial axis lengths can be found in "S2 Data", which is included in the supporting information for this study.

## 3.2. Growth parameters

After observation NJ and TL were not successfully overwintered. Measurements of the current year's growth of the six hazelnut populations that were successfully overwintered were taken after plant growth had ceased, and the results were shown in Table 6. The analysis showed that after the year's growth, MDJ had the highest plant height of 112.53 cm and NH2 had the largest ground diameter of 13.74 cm. There was a highly significant difference in plant height ($P < 0.01$). The plant height and ground diameter measurements are referred to "S3 Data", which is included in the supporting information for this study.

## 3.3. Nut economic traits

**3.3.1 Nut morphological characteristics.** Observations revealed that XK did not fruit. Table 7 presents the morphological characteristics of fruits from four provenance sources. NH2 had the largest nut width, length, thickness, weight, kernel weight, and hundred-grain weight among the populations, though its kernel percentage was low. NH1 had the highest

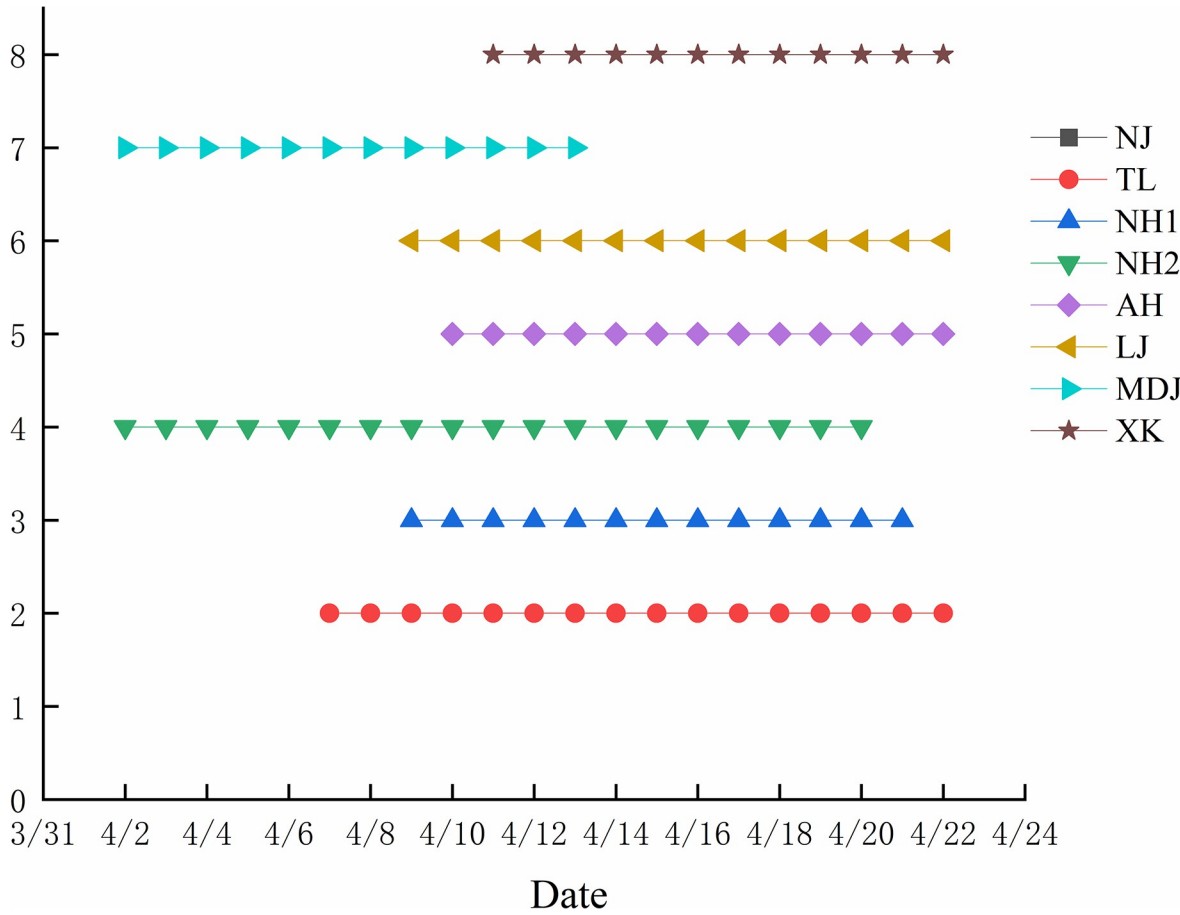

**Fig 2. Comparison of female inflorescence of different populations.** Note: NJ is not flowering.

kernel percentage at 86.53% and the largest shell thickness. ANOVA revealed significant differences in nut weight and kernel weight ($P < 0.05$) and highly significant differences in hundred-grain weight ($P < 0.01$) among hazelnut varieties. The shapes of hazelnut nuts, an important economic trait, vary among populations. According to the Specification for the Description of Hazelnut Germplasm Resources and Fig 4, NH1 and NH2 were oblate, while AH, LJ, and MDJ were round. For measurement data of hazelnut fruit, refer to "S4 Data" in supporting information.

**3.3.2 Correlation analysis of economic traits in nuts.**  The results of the correlation analysis on the economic traits of nuts of different hazelnut populations were shown in Fig 5, in which, the nut weight was significantly correlated with kernel weight, and highly significantly correlated with hundred-grain weight. Kernel weight was significantly correlated with nut length and kernel percentage. Hundred-grain weight was significantly correlated with nut length. This suggests that the nut length of different hazelnut populations was the one of important factor affecting their kernel weight and hundred-grain weight, thus indirectly affecting the kernel percentage.

### 3.4. Tree populations selection

**3.4.1 Principal component analysis (PCA).**  To evaluate the economic traits of the nuts of the test populations more comprehensively, 7 indicators of economic traits of nuts of 5

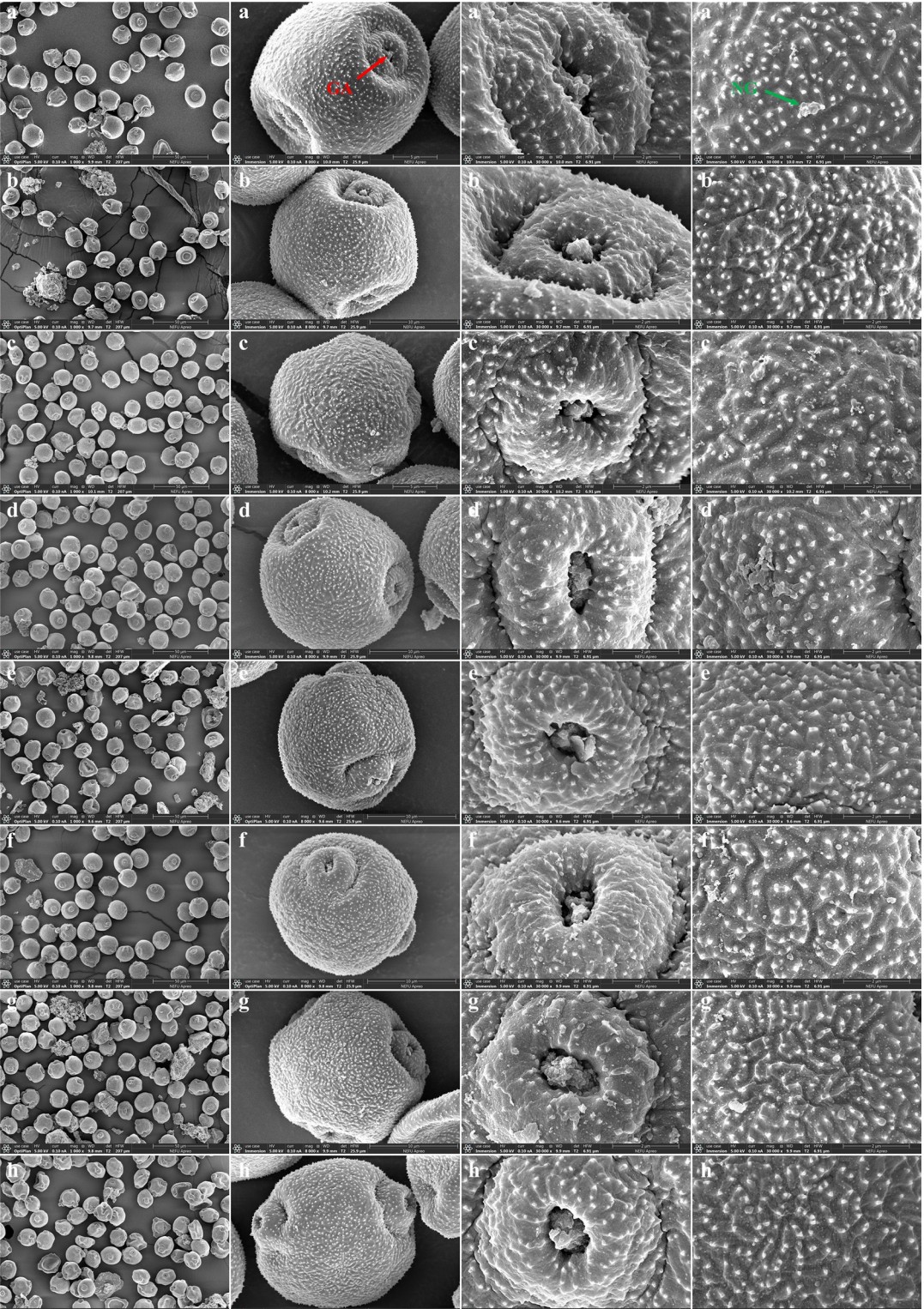

**Fig 3. Pollen morphology of each hazelnut populations.** Where a-h stand for NJ, TL, NH1, NH2, AH, LJ, MDJ, XK, respectively. Arrows in the figure: GA is Pollen germination apertures; NG is Nodular granule.

**Table 5. Comparison of morphological characteristics of pollen grains of different hazelnut.**

| Populations | Polar axis(μm) | Equatorial axis(μm) | Polar axis/Equatorial axis | Pollen shape | Germination aperture characteristics |
|---|---|---|---|---|---|
| NJ | 17.09±2.17A | 18.75±2.16A | 0.91 | Spherical | Invagination |
| TL | 14.80±1.17B | 16.80±1.05B | 0.88 | Nearly spherical | Invagination |
| NH1 | 15.19±0.78B | 16.49±1.06B | 0.92 | Spherical | Slight invagination |
| NH2 | 15.08±1.33B | 16.72±1.06B | 0.90 | Spherical | Invagination |
| AH | 14.02±0.98B | 16.02±0.85B | 0.88 | Nearly spherical | Slight invagination |
| LJ | 14.81±1.19B | 16.60±1.09B | 0.89 | Spherical | Invagination |
| MDJ | 14.70±0.85B | 16.28±1.00B | 0.90 | Spherical | Invagination |
| XK | 14.87±1.11B | 16.96±0.96B | 0.88 | Nearly spherical | Deep invagination |

Note: different capital letters indicate multiple comparisons at the 0.01 level.

**Table 6. Plant height and ground diameter of different hazelnut populations.**

| Populations | Plant height(cm) | Ground diameter(mm) |
|---|---|---|
| NH1 | 94.10±15.99bc | 9.30±2.49b |
| NH2 | 98.13±19.91bc | 13.74±12.19a |
| AH | 100.03±15.86b | 10.58±11.46ab |
| LJ | 98.48±11.28bc | 9.54±2.46b |
| MDJ | 112.53±21.17a | 10.11±2.46ab |
| XK | 90.20±16.43c | 9.16±2.70b |

Note: different lowercase letters indicate multiple comparisons at the 0.05 level.

hazelnut populations from 4 different provenances were selected for principal component analysis. The KMO value was 0.568, and the Bartlett's test of sphericity corresponded to $P < 0.01$, indicating that the selected indexes were suitable for principal components analysis, and the results of the analysis were shown in Table 8. Using an eigenvalue threshold greater than 1 as the standard, a total of two principal components were extracted. The variance contribution rate for the first principal component was 55.127%, representing 55.127% of the total trait information. The primary factors influencing the first principal component were nut thickness and nut length, which reflect the size of the nut. The variance contribution rate for the second principal component was 22.489%, with kernel percentage and kernel weight as the main factors, reflecting nut yield. Combined with Fig 6, it can be seen that the scatter of NH2 on the scatter plot of principal component analysis was distributed on one side, which was well distinguished from other populations.

**Table 7. Comparison of economic characteristics of different populations of hazelnut nuts.**

| Populations | Nut width (mm) | Nut length (mm) | Nut thickness (mm) | Nut weight (g) | Kernel weight (g) | Shell thickness (mm) | Hundred-grain weight(g) | Kernel percentage (%) |
|---|---|---|---|---|---|---|---|---|
| NH1 | 15.26±1.07c | 14.68±0.34a | 13.86±1.29b | 1.54±0.02b | 1.33±0.04a | 2.50±0.49a | 124.75±6.15b | 86.53±3.14a |
| NH2 | 16.51±0.37a | 15.35±0.43a | 15.24 ±0.48a | 1.80±0.08a | 1.40±0.19a | 2.06±0.35a | 157.76±10.74a | 77.50±6.73a |
| AH | 16.07±0.24ab | 14.70±0.40a | 14.21±0.58b | 1.58±0.19b | 1.25±0.07ab | 1.97±0.13a | 128.49±2.16b | 80.01±12.57a |
| LJ | 15.13±0.58c | 14.86±0.55a | 13.80±0.43b | 1.50±0.04b | 1.24±0.10ab | 2.36±0.12a | 132.20±5.47b | 82.69±6.92a |
| MDJ | 15.40±0.27bc | 14.62±0.13a | 13.95±0.11b | 1.44±0.09b | 1.07±0.04b | 2.20±0.10a | 129.34±1.30b | 74.56±6.39a |

Note: different lowercase letters indicate multiple comparisons at the 0.05 level. The NH2 shell was patterned.

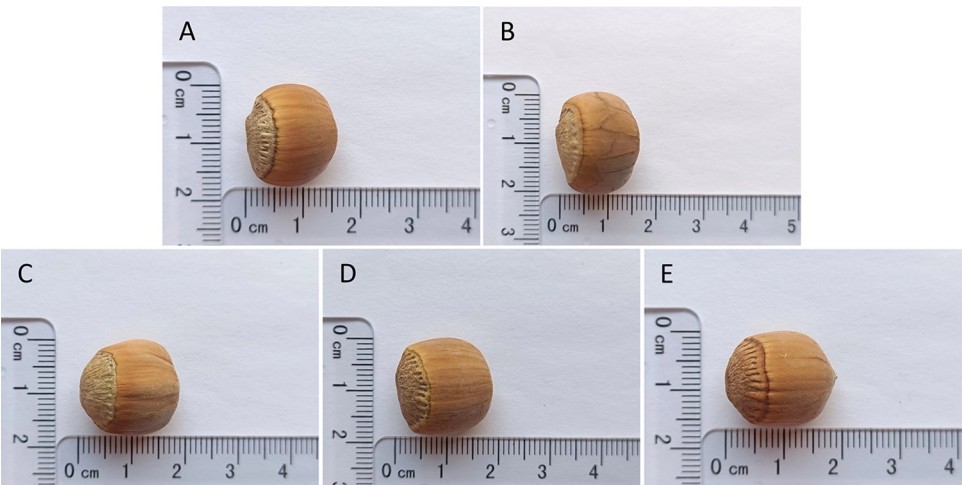

**Fig 4. Comparison of the appearance of different populations of hazelnut nuts.** A-E stand for NH1, NH2, AH, LJ, MDJ, respectively.

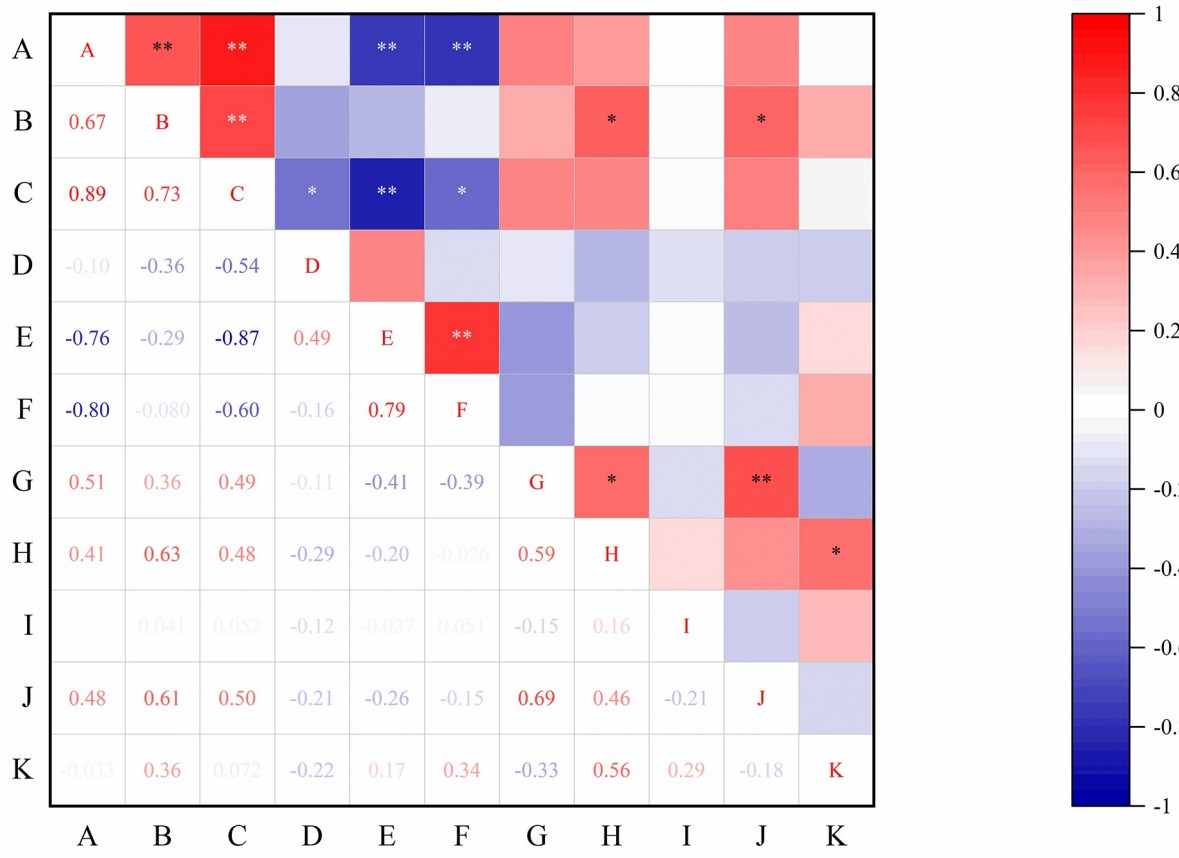

\* p⟨=0.05  \*\* p⟨=0.01

**Fig 5. Correlation analysis of morphological characters of different populations of hazelnut nuts.** "\*" and "\*\*" indicate significant correlation at the 0.05 level and 0.01 level, respectively. A, Nut Width; B, Nut Length; C, Nut Thickness; D, Nut Width/ Nut Thickness; E, Nut Length/ Nut Thickness; F, Nut Length/ Nut Width; G, Nut weight; H, Kernel weight; I, Shell thickness; J, Hundred-grain weight; K, kernel percentage.

**Table 8. Eigenvalues, variance contribution rates, and cumulative contribution rates.**

| Index | Component | |
|---|---|---|
| | X1 | X2 |
| Nut Width | 0.863 | -0.13 |
| Nut Length | 0.881 | 0.293 |
| Nut Thickness | 0.895 | -0.019 |
| Nut weight | 0.753 | -0.475 |
| Kernel weight | 0.777 | 0.458 |
| Hundred-grain weight | 0.79 | -0.34 |
| Kernel percentage | 0.138 | 1.016 |
| Eigenvalue | 4.136 | 1.687 |
| Variance contribution rate (%) | 55.127 | 22.489 |
| Cumulative contribution rate (%) | 55.127 | 77.616 |

**3.4.2 Comprehensive evaluation of principal components.** Based on the results of principal component analysis, the raw data of 7 nut economic traits of 5 hazelnut populations from 4 provenances were standardized and brought into 2 principal component score expressions

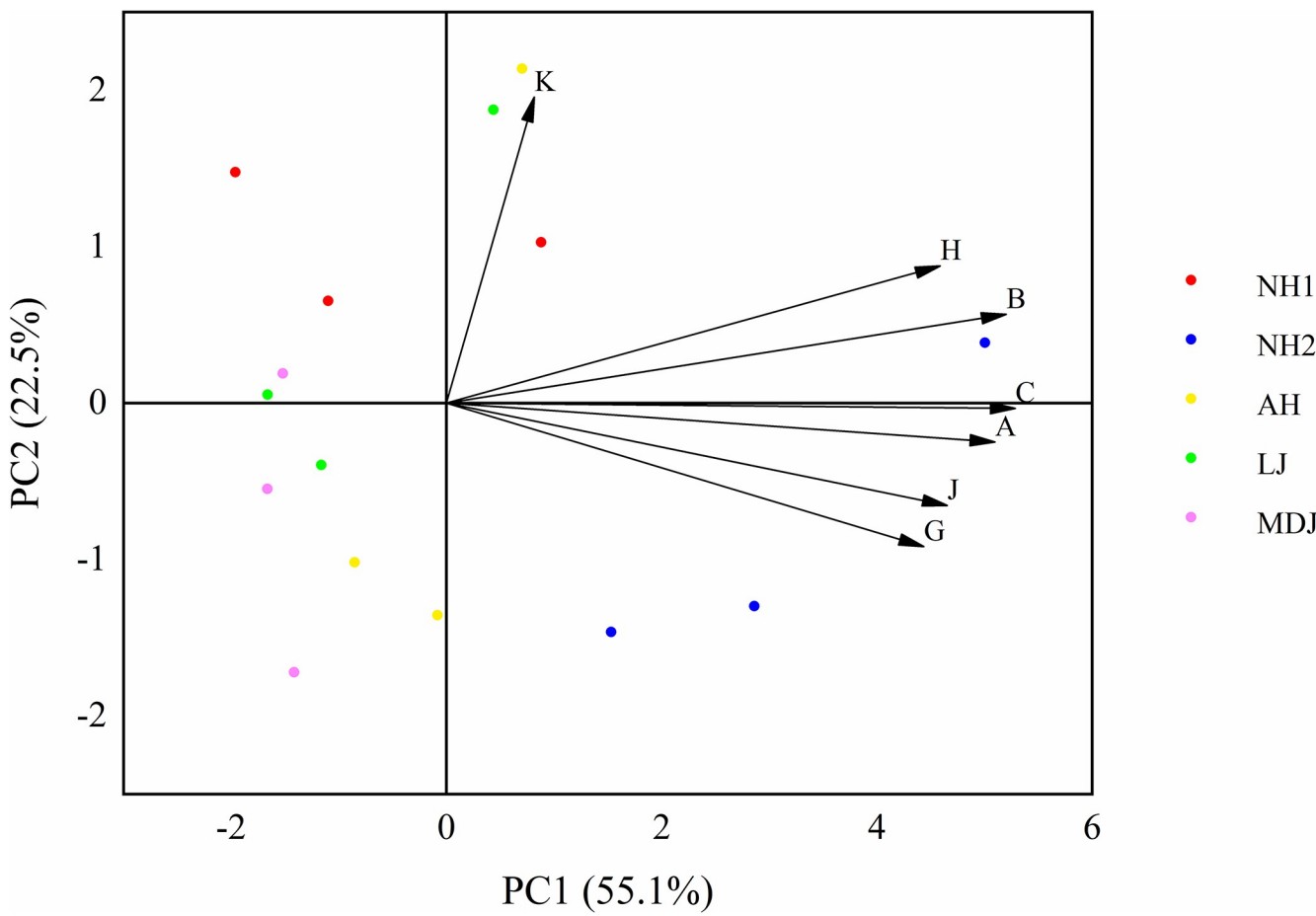

**Fig 6. Principal component analysis scatterplot.** A, Nut Width; B, Nut Length; C, Nut Thickness; G, Nut weight; H, Kernel weight; J, Hundred-grain weight; K, Kernel percentage.

**Table 9. Scores of each principal component score of different hazelnut populations and comprehensive ranking results.**

| Populations | F1 | F2 | F | Comprehensive ranking |
|---|---|---|---|---|
| NH1 | -2.427 | 1.571 | -1.269 | 3 |
| NH2 | 2.097 | -0.591 | 1.318 | 1 |
| AH | -1.640 | 0.212 | -1.103 | 2 |
| LJ | -2.863 | 0.795 | -1.803 | 4 |
| MDJ | -3.814 | -0.693 | -2.910 | 5 |

for calculation. The expressions for the scores of the 2 principal components and the comprehensive scores of the 5 populations for testing were as follows:

$$F_1 = 0.863X_1 + 0.881X_2 + 0.895X_3 + 0.753X_4 + 0.777X_5 + 0.79X_6 + 0.138X_7 \quad (3)$$

$$F_2 = -0.13X_1 + 0.293X_2 - 0.019X_3 - 0.475X_4 + 0.458X_5 - 0.34X_6 + 1.016X_7 \quad (4)$$

$$F = (0.551271F_1 + 0.22489F_2)/0.77616 \quad (5)$$

Where F1 and F2 denoted the scores of principal component 1 and principal component 2, F denoted the composite principal component score, and $X_1 \sim X_7$ denoted the values after standardization of the raw data of economic traits of different populations of nuts, respectively. The results of the comprehensive scores of the five hazelnut populations were shown in Table 9, and the comprehensive scores were NH2, AH, NH1, LJ, and MDJ in descending order. The comprehensive evaluation of all aspects showed that NH2 had the most excellent traits, and it was the wild hazelnut populations with excellent traits within the range of Northeast China.

## 3.5. Populations characteristics

**3.5.1 Comparison of plant morphology.** Comparison of the morphological characteristics of the plants of NH2 and NH1 was shown in Figs 7 and 8. The results showed that the first-year branches of both NH2 and NH1 were tender green and not lignified, the branchlets of NH2 were densely covered with stiff hirsute, and the branchlets of NH1 had sparse stiff hirsute. Second-year branches of both NH2 and NH1 were gray and woody, with NH2 densely covered with brown blotchy bumps and more branched than NH1. Leaves of both NH2 and NH1 were broadly obovate, with a triangular central cusp and a cordate base; leaf margins irregularly biserrate; petioles slender, densely covered with short hairs; lateral veins pinnate, 3–5 pairs, of which the leaf apices were notched in NH2, and the leaf apices were not notched in NH1. Overall, the nutritional content of the NH2 kernel is relatively high.

**3.5.2 Kernel nutrition.** The nutrient contents of the kernels of NH2 and NH1 were measured, including 17 amino acid contents, mineral element contents, soluble sugar contents, crude fat contents, crude protein contents and vitamin C contents, and the results were shown in Table 10. Comparison of the two populations can be obtained, per 100g of kernels, NH2 of the 17 amino acid content of Ala, Val *, IIe *, Leu *, His, Arg contents were more ("*" indicates the essential amino acids), the crude fat content than the NH1 of 5.7g, vitamin C content than the NH1 of 0.3g, NH2 of the mineral content were slightly less than the NH1, the rest of the nutrient content were less different. The test report on the nutritional content of hazelnut kernels by Qingdao Standard Testing Group Co., Ltd., please refer to "S1 File", has been included in the supporting information.

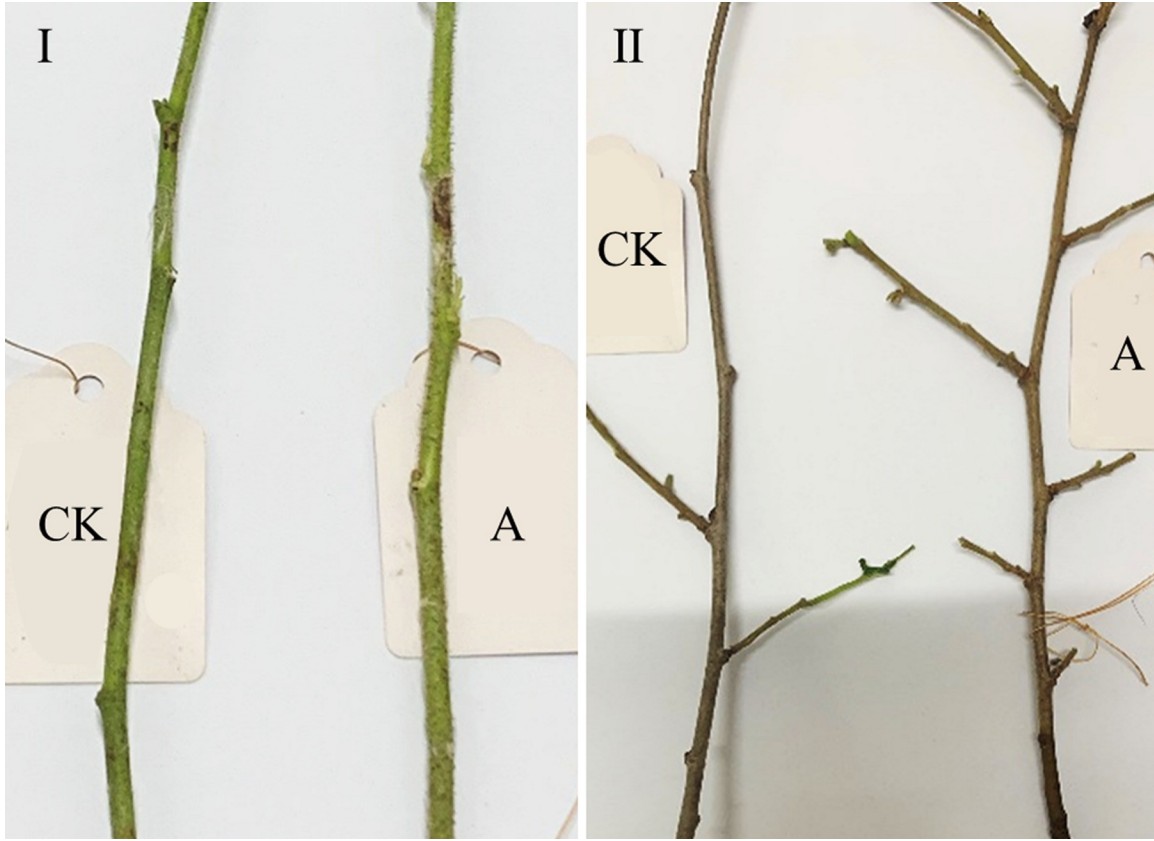

**Fig 7. Comparison of hazelnut branches.** I, First-year branches; II, Second-year branches; A, NH2; CK, NH1.

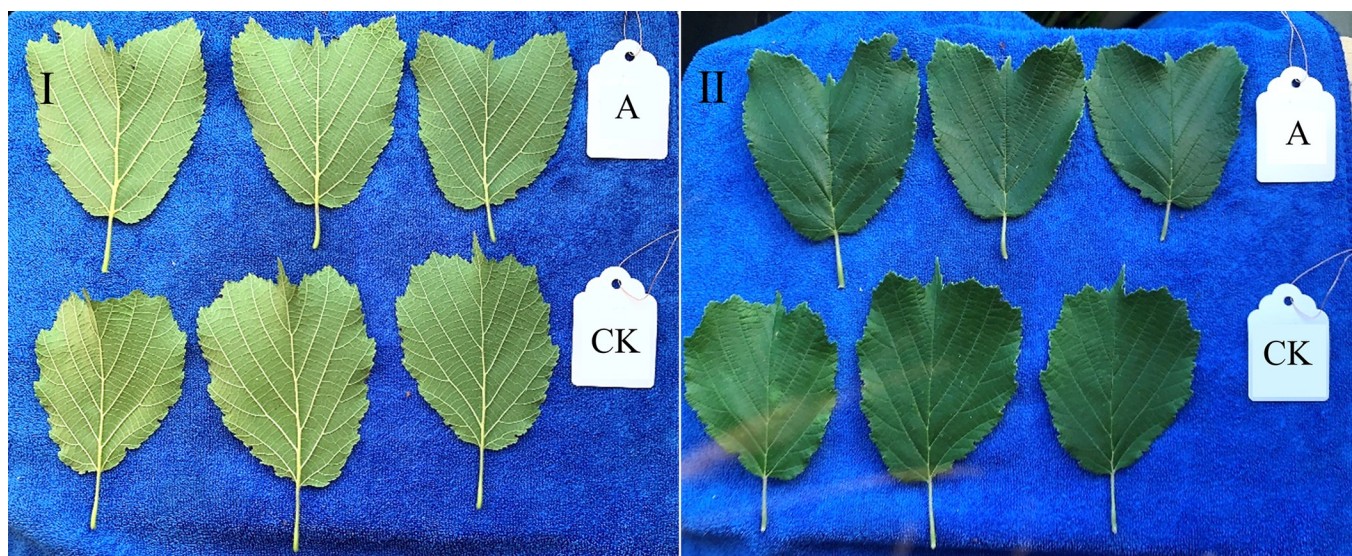

**Fig 8. Comparison of hazelnut leaves.** I, Back of the leaf; II, Leaf front; A, NH2; CK, NH1.

**Table 10. Comparison of nutrient composition of kernels.**

| Quality index | | Unit | NH2 | NH1 |
|---|---|---|---|---|
| 17 amino acids | Asp | g/100g | 1.68 | 1.71 |
| | Thr* | g/100g | 0.51 | 0.52 |
| | Ser | g/100g | 0.69 | 0.69 |
| | Glu | g/100g | 3.48 | 3.57 |
| | Gly | g/100g | 0.77 | 0.77 |
| | Ala | g/100g | 0.82 | 0.80 |
| | Cys | g/100g | 0.21 | 0.23 |
| | Val* | g/100g | 0.85 | 0.83 |
| | Met | g/100g | 0.13 | 0.15 |
| | IIe* | g/100g | 0.68 | 0.67 |
| | Leu* | g/100g | 1.16 | 1.15 |
| | Tyr | g/100g | 0.46 | 0.46 |
| | Phe* | g/100g | 0.75 | 0.77 |
| | Lys* | g/100g | 0.53 | 0.56 |
| | His | g/100g | 0.44 | 0.42 |
| | Arg | g/100g | 2.95 | 2.82 |
| | Pro | g/100g | 0.53 | 0.53 |
| Mineral element | K | mg/kg | $6.24 \times 10^3$ | $7.89 \times 10^3$ |
| | Ca | mg/kg | $2.07 \times 10^3$ | $2.49 \times 10^3$ |
| | Mg | mg/kg | $2.46 \times 10^3$ | $2.67 \times 10^3$ |
| | Fe | mg/kg | 48.3 | 57.4 |
| | Zn | mg/kg | 55.5 | 56.3 |
| Other Ingredient Content | Soluble sugar | % | 7.07 | 7.18 |
| | crude fat | g/100g | 51.6 | 45.9 |
| | Crude protein | g/100g | 24.7 | 25.6 |
| | VC | mg/kg | 1.53 | 1.23 |

Note: "*" indicates essential amino acids.

## 4. Discussion

Selection and breeding of superior wild hazelnut populations can lay the foundation for forest breeding work. To effectively develop and utilize wild hazelnut resources, it is necessary to carry out research on populations characteristics of some wild hazelnut populations. Populations characterization is a systematic study of the characteristics, traits, genetic basis, adaptability, and other aspects of a particular specie. Specifically, it includes research on external morphology, growth and development characteristics, yield, and quality, etc., to find high-yielding and high-quality superior populations. This study finally selected Nehe provenance of the striped *Corylus heterophylla* Fisch. as a superior populations through various screening, which laid the foundation for wild hazelnut breeding work.

Physiological florescence mainly refers to the flowering period of plants, and due to different environmental and genetic factors, the physical florescence of different populations will have some differences. Wang Wenbo [45], through the observation of phenology, especially the male and female florescence, was interested in understanding and grasping the flowering pattern of hazelnut, which in turn lays the foundation for the research of hybridization and cultivation of stock breeding. Wang Qi [46] studied the flowering and fruiting biology of five populations of *Corylus heterophylla* ×*Corylus avellana*, in which the results of phenological

observations were similar to those of our study, and the female florescence could meet the male florescence, but the initial flowering stage of male florescence of all populations in our study was the same. Li Ning et al. [47]. observed the phenology of three major hazelnut populations, and their results showed that the florescence and phenology of female flowers of different populations of hazelnut were similar, and the male florescence varied greatly, while the results of our study were the opposite, the florescence and phenology of male flowers of different populations of hazelnut were similar, and the time of the female florescence was different. The observation of male and female florescence is of great significance in grasping the flowering pattern of hazelnut plants, and then for the study of fruiting.

The shape, size and surface texture of pollen grains varied among populations, superior lines and strains, which can be used as a basis for classification and kinship studies of Hazels [48]. N. Nikolaieva et al. investigated the morphology of pollen grains of seven populations, which were sub compacted, compressed, or compressed globular and contained three germination pores [49]. Li Jingjing et al. used 14 different populations of hazelnut pollen as materials and observed the shape, size and surface texture of the pollen using a scanning electron microscope [50]. The results showed that the pollen grains of their test materials were nearly oblate spherical in shape with three germination pores, which were the same as the results of our study, but the texture differed from that of our study, which was granular texture, and that of the results of the observations made by Li Jingjing et al. was spinelike texture.

As winter temperatures are low in Northeast China, overwintering conditions and growth are important indicators for assessing the adaptability of populations. Zhang Caihong et al. observed the growth of a 3-year-old *Corylus heterophylla* ×*Corylus avellana*, Liaozhen3, in different regions of Shanxi and analyzed its adaptability [51]. Lv Mengyan et al. analyzed the morphological characters, growth characteristics, nut characteristics, and suitability range of two populations, Dawei and Liaozhen7, through the introductory cultivation experiment, and the results showed that these two populations were suitable for popularization and cultivation in Jilin [52]. In our experiment, the overwintering and growth of eight populations were observed, and the results showed that the overwintering of NJ and TL was poor, while the rest of the populations overwintered successfully. This study observed only one year of growth due to time constraints. However, in some research, a one-year growth index holds significant reference value [53]. Future studies should aim for long-term data collection to enhance the stability of growth index analysis results.

As an economic tree species, hazelnut nut quality is an important criterion for selecting the superior wild hazelnut. Gu Meiying et al. [54] used five populations of hazelnut as materials and comparatively analyzed the nut morphological characteristics of different populations of hazelnut, and their results analyzed that the nut length was an important factor influencing the kernel weight and hundred-grain weight, and the results obtained in our study by analyzing the correlation were the same as the conclusions of Gu Meiying et al. Cui Lin et al. [55]. measured and comprehensively evaluated the growth trait indexes of 12 provenances, and evaluated and screened the superior hazelnut provenance by using the method of principal component analysis, and the comprehensive ranking of Nehe provenance in the scoring results was the first one, and at the same time the growth trait performance of Nehe provenance was better, which was similar to the comprehensive evaluation results of our study, and Nehe provenance of the striped *Corylus heterophylla* Fisch. were the superior populations screened in the end.

There is some genetic variation in different populations of hazelnut nuts. Previous studies have shown that the nutrient composition and content of hazelnuts vary considerably among different populations or cultivars [56, 57]. The protein content of hazelnut kernels ranged from 12.6 to 25.9 g/100 g. The protein content of hazelnut kernels determined in

our study ranged from 24.7 to 25.6 g/100 g, which was consistent with the results of the above studies. Among the hazelnut populations studied by Jiang J [58], all populations of hazelnuts had relatively high Glu and Arg contents, and the contents of the 17 amino acids determined in our study were also relatively high in terms of Glu and Arg. Li Hongli et al. [59]. analyzed and evaluated the nut quality traits and nutrient composition of *Corylus heterophylla* Fisch. from nine different provenance in Heilongjiang Province, Northeast China, and the results showed that *Corylus heterophylla* Fisch. were characterized by high content of protein, fat, potassium, and zinc, which were like the results of the present experiment, and the results of our study showed that the crude fat content and vitamin C content of NH2 were higher than those of NH1.

This study conducted a comprehensive evaluation to identify superior provenances, although certain limitations remain. The analysis of variance (ANOVA) employed in this experiment included both within-group and between-group variances to assess differences in various hazelnut characteristics—such as florescence, pollen sub-microstructure, growth parameters, and economic traits—among different provenance. Intergroup variation primarily resulted from genetic differences among geographic provenance sources, which were statistically represented as variation between groups. These genetic differences significantly impacted hazelnut performance, providing guidance for the selection of superior provenance sources and improved varieties. In contrast, intra-group variation was predominantly attributed to individual differences within the same geographic provenance, likely due to environmental factors. Within-group variation indicates that even within the same geographic provenance, individual differences affect hazelnut characteristics. This suggests that individual differences must be considered in variety selection and cultivation management.

## 5. Conclusions

Taken together, our study successfully investigated eight hazelnut populations from seven provenances' flowering biology, kernel percentage, and the economic attributes of the produced nuts. By employing a rigorous principal component analysis, we finally identified elite populations, NH2. This work contributes to the hazelnut breeding realm by offering superior germplasm resources. Moreover, it plays an important role in the context of selecting high-quality hazelnut varieties, facilitating the identification, utilization, and conservation of wild hazelnut resources.

## Supporting information

**S1 Data. Male inflorescences data.**
(XLSX)

**S2 Data. Pollen submicroscopic structure data.**
(XLSX)

**S3 Data. Growth parameters data.**
(XLSX)

**S4 Data. Fruit economic attributes data.**
(XLSX)

**S1 File. Fruit nutrition.**
(PDF)

## Acknowledgments

We express our gratitude to Liguo Wu for the provenance trial site at the Cuhai Forest Farm. We thank Qingdao Stander Standard Testing Co. for providing the nutrient content test report of the kernel. We thank the Forest Breeding Laboratory of Qiqihar University. We appreciate the Hazel germplasm resource description specification provided by the China Crop Germplasm Resource Platform.

## Author Contributions

**Conceptualization:** Qiwen Yuan, Chunyu Guan.

**Data curation:** Qiwen Yuan.

**Formal analysis:** Qiwen Yuan, Dongyang Zhang.

**Investigation:** Qiwen Yuan, Yang Chen, Dongyang Zhang, Siyu Yang, Minghui Yang, Xuesong Zhu.

**Methodology:** Qiwen Yuan, Chunyu Guan.

**Software:** Qiwen Yuan, Yang Chen.

**Validation:** Qiwen Yuan, Yang Chen.

**Writing – review & editing:** Qiwen Yuan, Chunyu Guan.

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
