## [Decision Letter · Decision Letter 0]

24 Jul 2024

PONE-D-24-23349Characteristics of wild hazelnut species in northeast China and selection of superior provenances：a provenance trial.PLOS ONE

Dear Dr. Guan,

Thank you for submitting your manuscript to PLOS ONE. After careful consideration, we feel that it has merit but does not fully meet PLOS ONE’s publication criteria as it currently stands. Therefore, we invite you to submit a revised version of the manuscript that addresses the points raised during the review process.

We look forward to receiving your revised manuscript.

Kind regards,

Ali Akbar Ghasemi-Soloklui, Ph.D

Academic Editor

PLOS ONE

Journal Requirements:

This research was funded by Basic Research Fees of Universities in Heilongjiang Province, China (145309628)(https://kygl.webvpn.qqhru.edu.cn/userAction!do_casLogin.action). 

5. Please amend either the title on the online submission form (via Edit Submission) or the title in the manuscript so that they are identical.

6. We note that Figure 1 in your submission contain map images which may be copyrighted. All PLOS content is published under the Creative Commons Attribution License (CC BY 4.0), which means that the manuscript, images, and Supporting Information files will be freely available online, and any third party is permitted to access, download, copy, distribute, and use these materials in any way, even commercially, with proper attribution. For these reasons, we cannot publish previously copyrighted maps or satellite images created using proprietary data, such as Google software (Google Maps, Street View, and Earth). For more information, see our copyright guidelines: http://journals.plos.org/plosone/s/licenses-and-copyright.

We require you to either present written permission from the copyright holder to publish these figures specifically under the CC BY 4.0 license, or remove the figures from your submission:

7. Please remove your figures from within your manuscript file, leaving only the individual TIFF/EPS image files, uploaded separately. These will be automatically included in the reviewers’ PDF.

Reviewers' comments:

Reviewer's Responses to Questions

**Comments to the Author**

1. Is the manuscript technically sound, and do the data support the conclusions?

Reviewer #1: Yes

Reviewer #2: Partly

Reviewer #3: Yes

2. Has the statistical analysis been performed appropriately and rigorously? 

Reviewer #1: Yes

Reviewer #2: Yes

Reviewer #3: Yes

3. Have the authors made all data underlying the findings in their manuscript fully available?

Reviewer #1: Yes

Reviewer #2: Yes

Reviewer #3: Yes

4. Is the manuscript presented in an intelligible fashion and written in standard English?

Reviewer #1: Yes

Reviewer #2: No

Reviewer #3: No

5. Review Comments to the Author

**Reviewer #1: **The research paper focuses on the characterization and evaluation of hazelnut species in Northeast China based on their growth, nut economic traits, and species selection criteria. Through meticulous measurements and statistical analyses, the study highlights the importance of assessing multiple economic traits to determine tree excellence, utilizing tools like Principal Component Analysis for comprehensive evaluations. Strengths of the paper include its detailed methodology, rigorous data collection and analysis techniques, and the incorporation of both morphometric and nutrient content analyses to identify target species with exceptional traits. Additionally, the paper emphasizes the significance of overwintering success and nut economic characteristics in selecting superior hazelnut species, providing valuable insights for future breeding and cultivation efforts in the region.

Shortcomings of the abstract of the paper include:

1. Lack of clarity on the research methodology: The abstract do not provide detailed information on the specific methodology used in the provenance trials and genetic analysis of the hazelnut species. Without a clear explanation of the experimental design, data collection methods, and statistical analysis, it is difficult for readers to assess the reliability and validity of the study's findings.

Shortcomings of the Introduction of the paper include:

1. Limited context and background information: While the introduction briefly touches upon the ecological and economic significance of wild hazelnut species in China, it lacks a comprehensive review of existing literature on hazelnut or other nut trees breeding, genetics, and cultivation practices. Providing more context and background information would help situate the study within the broader research landscape and highlight the novelty and significance of the findings. For example, the author may use the following papers:

10.3390/plants10112234

10.1016/j.scienta.2020.109369

10.1007/978-3-030-23112-5_11

10.1016/j.scienta.2022.110885

10.17660/ActaHortic.2016.1139.18

10.17660/ActaHortic.2021.1318.37

10.1007/978-981-19-9497-5_4

https://www.researchgate.net/publication/379666419_Targeted_Breeding_Innovations_in_Nut_Tree_Cultivars

2. Absence of comparative analysis: The introduction emphasizes the geographical variations and unique characteristics of the different provenances of wild hazelnut species in Northeast China. However, there is a lack of comparative analysis or discussion on how these variations may impact the hazelnuts' overall productivity, adaptability, and market value. A more thorough comparative analysis would enhance the depth of the study and provide stronger insights for potential breeding and cultivation strategies. For example, the author may benefit from the following papers:

http://dx.doi.org/10.1016/j.scienta.2024.113275

10.1093/jxb/erz467

10.22059/ijhst.2020.299930.352

10.1093/hr/uhac124

10.17660/ActaHortic.2015.1074.18

4. Limited discussion on sustainability and future implications: While the introduction briefly mentions the industrial and medicinal benefits of hazelnuts, there is a lack of discussion on the sustainability aspects of hazelnut cultivation and the potential long-term implications of the study's findings. Including a section on the environmental sustainability, socio-economic impact, and future research directions would enrich the paper and demonstrate a broader perspective on the importance of wild hazelnut biodiversity conservation and utilization.

The "Materials and Methods" section of the research paper contains valuable details about the procedures and techniques utilized in the study. However, there are a few shortcomings that could be addressed to enhance the clarity and rigor of the methodology:

1. Lack of detail on sampling methodology: The section briefly mentions the collection of hazelnut seeds from various provenances within Northeast China, but it does not provide specific details on the sampling strategy employed. Information on how the seeds were collected, the criteria used for selecting specific provenances, and the rationale behind the selection of the eight hazelnut species could strengthen the transparency and reproducibility of the study.

2. Insufficient information on experimental design: While the section mentions the establishment of provenance trial forests with a randomized complete block design and three repeats, it lacks explicit details on the allocation of treatments, the randomization process, and the specific variables measured in the study. Providing a more comprehensive description of the experimental design, including the control measures implemented, could aid in better understanding the reliability and validity of the results obtained.

3. Limited explanation of data analysis techniques: The section briefly mentions the use of statistical analysis tools such as principal component analysis (PCA) and Microsoft Excel for processing and analyzing the data. However, it would be beneficial to include more information on the specific statistical tests used, how the data were managed and cleaned, and any assumptions underlying the analyses. This would provide readers with insights into the robustness of the statistical findings and interpretations.

4. Lack of discussion on potential biases and limitations: The section does not address any potential biases or limitations that may have affected the study's outcomes. Acknowledging potential sources of bias, such as environmental variability, sampling errors, or measurement inconsistencies, could help contextualize the results and offer insights into the generalizability of the findings.

By addressing these shortcomings and providing more detailed information on sampling, experimental design, data analysis, and potential biases, the "Materials and Methods" section could be strengthened to ensure the robustness and reliability of the study's findings.

**Reviewer #2:** This research has investigated some characteristics of wild hazelnut species in northeastern China to selection of superior provenances. Many indicators have been measured. However, the results of this research are mostly of local use and may not be interesting for readers from other parts of the world. In addition, the evaluation of genetic diversity based on morphological characters alone without the use of molecular techniques belongs to the last decade, and nowadays new techniques are used to evaluate genetic diversity. It is not clear how to choose the species.

Were they selected from the population or were the characteristics of these species already known? If there was a population, how was the selection of the population? What was the size of the population (how many hectares). One-year measurement data is not reliable and should be repeated at least two or three years for morphological and phenological indicators.

This manuscript is recommended for publication in local journals.

**Reviewer #3:** It is an interesting study, evaluating valuable genetic resources of hazelnut in China. It contain quite good mod of data presentation and quality macro and micro pictures, but sufferers some weak points which needs to be considered;

-Some sections specially materials and methods as well as the results are too lengthy and needs to be summarized.

-the overall English is good but still needs to be improved preferably by a native speaker.

-As mentioned in introduction as well as in table 1, the all plant materials belongs to two distinct species, this is while authors speak about 8 distinct species!! it seems they need to name the plant materials belonged to one species as populations, ecotype etc.

-Some terms are not selected properly; for example growth condition (instead you can use growth attributes/ or parameters), long in table 2 (instead use length), flowering length in table 3 and 4 (use flowering period/ or duration), percent kernel (use kernel percentage) etc.

-The line for NJ is absent in Fig 2 & 3.

-For mean comparison in tables use same order of alphabet letters from highest value to lowest value (a/ or A for highest value).

6. PLOS authors have the option to publish the peer review history of their article (what does this mean?). If published, this will include your full peer review and any attached files.

Reviewer #1: No

Reviewer #2: No

Reviewer #3: No

---

## [Author Response · Author response to Decision Letter 0]

11 Sep 2024

Dear Editors and Reviewers,

Thank you for your letter and for the reviewer’s comments concerning our manuscript entitled “Characteristics of wild hazelnut populations in northeast China and selection of superior provenances” (ID: PONE-D-24-23349).Those comments are all valuable and very helpful for revising and improving our paper, as well as the important guiding significance to our researches. We have studied comments carefully and have made correction which we hope meet with approval. Furthermore, we would like to show the details as follows:

Academic Editor：

Please ensure that your manuscript meets PLOS ONE's style requirements, including those for file naming. The PLOS ONE style templates can be found at https://journals.plos.org/plosone/s/file?id=wjVg/PLOSOne_formatting_sample_main_body.pdf and https://journals.plos.org/plosone/s/file?id=ba62/PLOSOne_formatting_sample_title_authors_affiliations.pdf

The author’s answer:

We have read the style templates carefully and have revised the manuscript to PLOS ONE's style, which we hope will meet your requirements. If there are any questions, please point them out and we will make corrections soon.

This research was funded by Basic Research Fees of Universities in Heilongjiang Province, China (145309628) (https://kygl.webvpn.qqhru.edu.cn/userAction!do_casLogin.action). 

The author’s answer:

Thank you for bringing that up. This research was funded by the Basic Research Fees of Universities in Heilongjiang Province, China (145309628), Graduate Innovation Project of Qiqihar University (X202310232032), Graduate Innovation Project of Qiqihar University (X202310232040).What role the funders took in the study: All funders provided financial support.

The author’s answer:

Thank you for bringing that up. We submit the minimal data set on the supporting information that contains the values behind the means, standard deviations, the values used to build graphs and the points extracted from images for analysis.

The author’s answer:

Thank you for your advice. We decide on a data sharing plan before acceptance.

5. Please amend either the title on the online submission form (via Edit Submission) or the title in the manuscript so that they are identical.

The author’s answer:

Thank you for pointing this out. We have amended the title on the online submission.

6. We note that Figure 1 in your submission contain map images which may be copyrighted. All PLOS content is published under the Creative Commons Attribution License (CC BY 4.0), which means that the manuscript, images, and Supporting Information files will be freely available online, and any third party is permitted to access, download, copy, distribute, and use these materials in any way, even commercially, with proper attribution. For these reasons, we cannot publish previously copyrighted maps or satellite images created using proprietary data, such as Google software (Google Maps, Street View, and Earth). For more information, see our copyright guidelines: http://journals.plos.org/plosone/s/licenses-and-copyright.

We require you to either present written permission from the copyright holder to publish these figures specifically under the CC BY 4.0 license, or remove the figures from your submission:

The author’s answer:

We have tried our best to obtain permission from the original copyright holders, but to no avail. We have therefore decided to remove the figure. We have also changed the numbering of the figures in the manuscript.

7. Please remove your figures from within your manuscript file, leaving only the individual TIFF/EPS image files, uploaded separately. These will be automatically included in the reviewers’ PDF.

The author’s answer:

Thank you for bringing that up. We have removed all figures from within manuscript file.

Reviewer #1:

Shortcomings of the abstract of the paper：

1. Lack of clarity on the research methodology: The abstract do not provide detailed information on the specific methodology used in the provenance trials and genetic analysis of the hazelnut species. Without a clear explanation of the experimental design, data collection methods, and statistical analysis, it is difficult for readers to assess the reliability and validity of the study's findings.

The author’s answer:

We thank the reviewer for pointing this out. We have provided detailed information on the specific methodology used in the provenance trials and genetic analysis of the hazelnut species. We have changed [This study addresses this knowledge gap by initiating provenance trials on eight distinctive wild hazelnut taxa, sourced from a broad array of seven different locations within Northeast China. The trials meticulously explored the trees' flowering biology, growth parameters, and the economic attributes of the produced nuts.] to [To develop and utilize high-quality wild hazelnut resources, this study selected eight populations of wild hazelnuts from seven different provenances in Heilongjiang Province, China. Natural hybrid seeds of eight populations were sown in the Chohai Forest Farm in Longjiang County, Heilongjiang Province, in 2018. In April 2020, two-year-old seedlings were used to establish a provenance trial forest, thereby initiating the provenance trial. Growth parameters were measured using electronic calipers, and pollen characteristics were observed under an electron microscope. The trials meticulously explored the trees' flowering biology, growth parameters, and the economic attributes of the produced nuts. Principal component analysis was employed to comprehensively assess differences among the wild hazelnut populations from various provenances, aiming to identify superior wild hazelnut provenances.] (page 1, line 4).

Shortcomings of the Introduction of the paper：

1. Limited context and background information: While the introduction briefly touches upon the ecological and economic significance of wild hazelnut species in China, it lacks a comprehensive review of existing literature on hazelnut or other nut trees breeding, genetics, and cultivation practices. Providing more context and background information would help situate the study within the broader research landscape and highlight the novelty and significance of the findings.

The author’s answer:

Thank you for your suggestions! The references you suggested are benefit for us. We have read all references and learn a lot. In response to your suggestions, we have included content on the breeding, genetics, and cultivation practices of hazelnuts and other nut trees to provide a more comprehensive. Please see page 3 of the revised manuscript, line 47-63. We hope that it is now clearer.

2. Absence of comparative analysis: The introduction emphasizes the geographical variations and unique characteristics of the different provenances of wild hazelnut species in Northeast China. However, there is a lack of comparative analysis or discussion on how these variations may impact the hazelnuts' overall productivity, adaptability, and market value. A more thorough comparative analysis would enhance the depth of the study and provide stronger insights for potential breeding and cultivation strategies.

The author’s answer:

Thank you for this valuable comments. We have read all references and learn a lot. As suggested by the reviewer, we have added content on how these variations may impact the hazelnuts' overall productivity, adaptability, and market value. Please see page 5 of the revised manuscript, line 93-101. 

3. Limited discussion on sustainability and future implications: While the introduction briefly mentions the industrial and medicinal benefits of hazelnuts, there is a lack of discussion on the sustainability aspects of hazelnut cultivation and the potential long-term implications of the study's findings. Including a section on the environmental sustainability, socio-economic impact, and future research directions would enrich the paper and demonstrate a broader perspective on the importance of wild hazelnut biodiversity conservation and utilization.

The author’s answer:

Thank you for highlighting this gap in our manuscript. We have added the suggested content to the manuscript on page 2 of the revised manuscript, line 42-46.

Shortcomings of the Materials and Methods of the paper:

1. Lack of detail on sampling methodology: The section briefly mentions the collection of hazelnut seeds from various provenances within Northeast China, but it does not provide specific details on the sampling strategy employed. Information on how the seeds were collected, the criteria used for selecting specific provenances, and the rationale behind the selection of the eight hazelnut species could strengthen the transparency and reproducibility of the study.

The author’s answer:

We think this is an excellent suggestion. We provide specific details on the sampling strategy employed:

(1) How the seeds were collected: Thirty dominant trees were selected from each geographical source to serve as seed trees, and their natural hybrid seeds were collected (page 6, line 118-120).

(2) The criteria used for selecting specific provenances: For this experiment, seed trees were selected based on criteria that included the absence of severe pests and diseases and higher productivity (page 6, line 113-114).

(3) The rationale behind the selection of the eight hazelnut species：These species are only found in Heilongjiang Province and have some excellent traits. We have changed [Eight hazelnut species from seven different provenance sources within Northeast China were selected for this experiment] to [Eight hazelnut species with some excellent traits distributed in seven different geographic provenance sources within the Heilongjiang Province were selected] (page 6, line 115-116).

2. Insufficient information on experimental design: While the section mentions the establishment of provenance trial forests with a randomized complete block design and three repeats, it lacks explicit details on the allocation of treatments, the randomization process, and the specific variables measured in the study. Providing a more comprehensive description of the experimental design, including the control measures implemented, could aid in better understanding the reliability and validity of the results obtained.

The author’s answer:

Thank you for your valuable suggestions. We added a more comprehensive description of the experimental design. We have changed [Seedlings were nurtured in the Cuohai Forest Farm in Longjiang County to create provenance trial forests in April 2020 using 2-year old se

---

## [Decision Letter · Decision Letter 1]

30 Sep 2024

PONE-D-24-23349R1Characteristics of wild hazelnut populations in northeast China and selection of superior provenancesPLOS ONE

Dear Dr. Guan,

Thank you for submitting your manuscript to PLOS ONE. After careful consideration, we feel that it has merit but does not fully meet PLOS ONE’s publication criteria as it currently stands. Therefore, we invite you to submit a revised version of the manuscript that addresses the points raised during the review process. Please submit your revised manuscript by Nov 14 2024 11:59PM. If you will need more time than this to complete your revisions, please reply to this message or contact the journal office at plosone@plos.org. Please include the following items when submitting your revised manuscript:A rebuttal letter that responds to each point raised by the academic editor and reviewer(s). You should upload this letter as a separate file labeled 'Response to Reviewers'.A marked-up copy of your manuscript that highlights changes made to the original version. You should upload this as a separate file labeled 'Revised Manuscript with Track Changes'.An unmarked version of your revised paper without tracked changes. You should upload this as a separate file labeled 'Manuscript'.If applicable, we recommend that you deposit your laboratory protocols in protocols.io to enhance the reproducibility of your results. Protocols.io assigns your protocol its own identifier (DOI) so that it can be cited independently in the future. For instructions see: https://journals.plos.org/plosone/s/submission-guidelines#loc-laboratory-protocols. Additionally, PLOS ONE offers an option for publishing peer-reviewed Lab Protocol articles, which describe protocols hosted on protocols.io. Read more information on sharing protocols at https://plos.org/protocols?utm_medium=editorial-email&utm_source=authorletters&utm_campaign=protocols.

We look forward to receiving your revised manuscript.

Kind regards,

Ali Akbar Ghasemi-Soloklui, Ph.D

Academic Editor

PLOS ONE

**Journal Requirements:**

**Reviewers' comments:**

Reviewer's Responses to Questions

**Comments to the Author**

1. If the authors have adequately addressed your comments raised in a previous round of review and you feel that this manuscript is now acceptable for publication, you may indicate that here to bypass the “Comments to the Author” section, enter your conflict of interest statement in the “Confidential to Editor” section, and submit your "Accept" recommendation.

Reviewer #1: (No Response)

Reviewer #3: All comments have been addressed

2. Is the manuscript technically sound, and do the data support the conclusions?

Reviewer #1: Yes

Reviewer #3: Yes

3. Has the statistical analysis been performed appropriately and rigorously? 

Reviewer #1: Yes

Reviewer #3: Yes

4. Have the authors made all data underlying the findings in their manuscript fully available?

Reviewer #1: Yes

Reviewer #3: Yes

5. Is the manuscript presented in an intelligible fashion and written in standard English?

Reviewer #1: Yes

Reviewer #3: Yes

6. Review Comments to the Author

**Reviewer #1:** The authors appear to have addressed most of the points raised by the reviewers effectively:

1. Methodology Clarification: The authors added more detailed descriptions regarding the provenance trials and genetic analysis, addressing Reviewer #1's concern. The revised abstract provides a clear breakdown of the methodology, including seed collection, pollen observations, and principal component analysis.

2. Introduction Enhancements: The authors expanded the introduction to include background on hazelnut breeding, genetics, and cultivation practices, which Reviewer #1 highlighted as lacking. The inclusion of broader context regarding hazelnut resource exploitation and the sustainability of hazelnut cultivation was also added, as suggested.

3. Materials and Methods: The authors responded to the need for more detail on sampling methodology, such as how seeds were collected, the selection criteria for provenances, and the experimental design. They provided information on the sampling and experimental procedures, including statistical methods, randomization, and how variables were measured. This meets Reviewer #1’s request for greater transparency and strengthens the study's reproducibility.

4. Length of Sections: The authors made attempts to summarize the lengthier sections, specifically the materials and methods and results, as suggested by Reviewer #3. However, further streamlining may still be necessary, especially in the results section, which contains a great deal of detail.

5. Comparison of Nut Traits and Geographic Variation: The paper also addressed Reviewer #2's suggestion on comparative analysis of productivity, adaptability, and market value by examining geographic variations in hazelnut characteristics. This was done through a detailed principal component analysis.

Points Needing Further Attention:

1. Reference Formatting: As you pointed out, some of the newly added references do not include the journal names, page numbers, or volumes. The authors should standardize their references in line with the required citation style of PLOS ONE. Ensuring complete and consistent referencing, including missing elements (e.g., journal name, pages, volume), is critical for a high-quality academic manuscript.

2. Clarification on One-Year Data: Reviewer #2 questioned the reliability of one-year measurement data, advising a longer observational period. The authors justified their use of one-year data by referencing another publication, but they should be clearer about potential limitations and perhaps indicate plans for longer-term data collection in future studies to strengthen this defense.

3. Potential for Further Summarization: While the authors shortened some sections, there are still lengthy parts, particularly in the results section. More concise presentation could enhance the paper’s readability.

Recommendations for Authors:

- Ensure reference formatting is fully compliant with the journal’s style guidelines. Include missing journal names, pages, and volumes.

- Continue revising for conciseness, particularly in the results section. Summarizing some repetitive details may help the paper be more reader-friendly.

**Reviewer #3:** (No Response)

7. PLOS authors have the option to publish the peer review history of their article (what does this mean?). If published, this will include your full peer review and any attached files.

Reviewer #1: No

Reviewer #3: No

---

## [Author Response · Author response to Decision Letter 1]

26 Oct 2024

Dear Editors and Reviewers,

Thank you very much for your comments and professional advice concerning our manuscript entitled “Characteristics of Wild Hazelnut Populations in Northeast China and Selection of Superior Provenances” (ID: PONE-D-24-23349R2). Your comments are valuable and very helpful for revising and improving the academic rigor of our article, as well as providing significant guidance for our research. We have studied the comments carefully and have made corrections, which we hope will meet with your approval. Furthermore, we would like to present the details as follows:

Academic Editor：

The author’s answer:

Thank you for your comments. We have revised our references in accordance with the journal's reference format requirements. The following are the details of the revision of the reference (The following numbers are reference numbers, with changes highlighted in red in the revised manuscript):

[2]: We added the number of pages. From[Brown JA, Beatty GE, Montgomery WI, Provan J. Broad-scale genetic homogeneity in natural populations of common hazel (Corylus avellana) in Ireland. Tree Genet Genomes. 2016;12(6). doi: ARTN 12210.1007/s11295-016-1079-7. PubMed PMID: WOS:000397238800022.] to [Brown JA, Beatty GE, Montgomery WI, Provan J. Broad-scale genetic homogeneity in natural populations of common hazel (Corylus avellana) in Ireland. Tree Genet Genomes. 2016;12(6):122. doi: 10.1007/s11295-016-1079-7. PubMed PMID: WOS:000397238800022].

[5]: We made the changes according to the requirements of the journal reference format. From[Zong JW. Study on Population Genetic Diversity of three Chinese Corylus Species and the Phylogentic Relationship of Corylus Species [PhD]: Chinese Academy of Forestry; 2017] to [Zong JW. Study on Population Genetic Diversity of three Chinese Corylus Species and the Phylogentic Relationship of Corylus Species. PhD. Thesis, Chinese Academy of Forestry; 2017. Available from: https://kns.cnki.net/kcms2/article/abstract?v=_W1AupcyYgZJYRJtAWj7Rty7HrHEdpbxgn9Qg1m4Msdu9fvGGxf9aUbhf2a3hQcSgkSxk4JKLgzHWheDOdXcswqGUsQHPO-ao1virqxCWge0oY-54IJw4MYaJ3DkWGTFAmDqHWWm1FhHUVd6nhN8dLBylgj1YZ4t5VzU51_e_NBlxQAHapTAcZPaD_WUuDDDfWXnsjfLcyI=&uniplatform=NZKPT&language=CHS].

[12]: We added DOI. From[Luo F, Fei XQ, Tang FB, Li XL. Simultaneous determination of paclitaxel in hazelnut by HPLC-MS/MS. Forest Research. 2011;24:779-83] to [Luo F, Fei XQ, Tang FB, Li XL. Simultaneous determination of paclitaxel in hazelnut by HPLC-MS/MS. Forest Research. 2011;24:779-83. doi: https://doi.org/10.1016/j.heliyon.2023.e13675].

[17]: We made the changes in accordance with the format requirements of the reference "Book chapters" in the journal. From[Vahdati K, Sheikhi A, Arab MM, Sarikhani S, Habibi A, Ataee H. Cultivars and Genetic Improvement. In: Mir MM, Rehman MU, Iqbal U, Mir SA, editors. Temperate Nuts. Singapore: Springer Nature Singapore; 2023. p. 79-111] to [Vahdati K, Sheikhi A, Arab MM, Sarikhani S, Habibi A, Ataee H. Cultivars and Genetic Improvement. In: Mir MM, Rehman MU, Iqbal U, Mir SA, editors. Temperate Nuts. Singapore: Springer Nature Singapore; 2023. pp. 79-111].

[18]: We have added journal names, volumes, pages and DOI. From[Vahdati K, Khorami SS, editors. The past, present and future of walnut genetic improvement and propagation2021: International Society for Horticultural Science (ISHS), Leuven, Belgium] to [Vahdati K, Khorami SS, editors. The past, present and future of walnut genetic improvement and propagation. Acta Hortic. 1318: 251-258. doi: https://doi.org/10.17660/ActaHortic.2021.1318.37].

[19]: We made the changes in accordance with the format requirements of the reference "Book chapters" in the journal. From[Vahdati K, Arab M, Sarikhani S, Sadat-Hosseini M, Leslie C, Brown P. Advances in Persian Walnut (Juglans regia L.) Breeding Strategies. 2019. p. 401-72] to [Vahdati K, Arab M, Sarikhani S, Sadat-Hosseini M, Leslie C, Brown P. In: Advances in Persian Walnut (Juglans regia L.) Breeding Strategies. Al-Khayri, J., Jain, S., Johnson, D. (eds) Advances in Plant Breeding Strategies: Nut and Beverage Crops. Springer, Cham; 2019. pp. 401-72].

[25]: We added pages. From[Zhao T, Ma W, Yang Z, Liang L, Chen X, Wang G, et al. A chromosome-level reference genome of the hazelnut, Corylus heterophylla Fisch. Gigascience. 2021;10(4). Epub 2021/04/20. doi: 10.1093/gigascience/giab027. PubMed PMID: 33871007; PubMed Central PMCID: PMCPMC8054262] to [Zhao T, Ma W, Yang Z, Liang L, Chen X, Wang G, et al. A chromosome-level reference genome of the hazelnut, Corylus heterophylla Fisch. Gigascience. 2021;10(4): 1-9 Epub 2021/04/20. doi: 10.1093/gigascience/giab027. PubMed PMID: 33871007; PubMed Central PMCID: PMCPMC8054262].

[29]: We have added journal names, volumes, pages and DOI. From[Vahdati K, Mohseniazar M, editors. Early bearing genotypes of walnut: a suitable material for breeding and high density orchards2016: International Society for Horticultural Science (ISHS), Leuven, Belgium] to [Vahdati K, Mohseniazar M, editors. Early bearing genotypes of walnut: a suitable material for breeding and high density orchards. Acta Hortic. 1139: 101-106. doi: https://doi.org/10.17660/ActaHortic.2016.1139.18].

[30]: We have added journal names, volumes, pages and DOI. From[Vahdati K, Karimi R, Ershadi A, editors. GENETIC STRUCTURE OF SOME WILD WALNUT POPULATIONS IN IRAN2015: International Society for Horticultural Science (ISHS), Leuven, Belgium] to [Vahdati K, Karimi R, Ershadi A, editors. GENETIC STRUCTURE OF SOME WILD WALNUT POPULATIONS IN IRAN. Acta Hortic. 1074: 125-128. doi: https://doi.org/10.17660/ActaHortic.2015.1074.18]

[34]: We are sorry that we did not find the DOI of this reference. But we have provided the link here: https://kns.cnki.net/kcms2/article/abstract?v=_W1AupcyYgZACCJ7g5T4YcpuIJNx0YaDr-dpt2j_vUqw28MX81RkRe2ARt84fqqCapqeVWZKnPHbRwU22_l7vVtPkwdAXJy2TSWpUQxHh-JjQEWzHfpG2GaU-Ul5OAJ0suup9_2ie4FJ5iMF8yqTsKKCPnF8gff7uNQlT7gemRg0L4_nuRm1Qhnwt8v8FP_P&uniplatform=NZKPT&language=CHS.

[35]: We added DOI. From[Long ZY, Lu CH. Resourse distribution and development research progress of Corylus in Heilongjiang province. Forest By-Product and Speciality in China. 2005;(04):41-2] to [Long ZY, Lu CH. Resourse distribution and development research progress of Corylus in Heilongjiang province. Forest By-Product and Speciality in China. 2005;(04):41-42. doi: :10.13268/j.cnki.fbsic.2005.04.030].

[45]: We revised the references according to the requirements in the journal reference format "Masters' theses or doctoral dissertations". From[WANG WB. Florescence Phenology and Techniques for Promoting Blossom and Fruiting of Corylus spp. [Master]: Northeast Forestry University; 2002] to [WANG WB. Florescence Phenology and Techniques for Promoting Blossom and Fruiting of Corylus spp. M.Sc. Thesis, Northeast Forestry University; 2002. Available from: https://kns.cnki.net/kcms2/article/abstract?v=_W1AupcyYgY_l3KMaGPX-H9LOGYFEl9x2B7urTOoRP_AnWmCXaTeE62_AN5hQW80-UczQml1K-YQTt-PJ0v4o1xUwijY9mSxG8QejPDA4nqiGwt3NINHEOKLAzTfFX6E4LUrsMZ5wzQL6rWwKetwN-4GhsztS4oy0RfhaDb_vK9aADjXuTFKNIv-nNsuf9Pn&uniplatform=NZKPT&language=CHS].

[46]: We revised the references according to the requirements in the journal reference format "Masters' theses or doctoral dissertations". From[Wang Q. Study on the biological characteristics of flowering and fruiting of hazelnut hybrid [Master]: Xinjiang Agricultural University; 2022] to [Wang Q. Study on the biological characteristics of flowering and fruiting of hazelnut hybrid. M.Sc. Thesis, Xinjiang Agricultural University; 2022. Available from: https://kns.cnki.net/kcms2/article/abstract?v=_W1AupcyYgY5BVeZhTs5dAw8WvT_Qn9jb8BkPiDd7qmLnqxr8oCR83NLX3-DuCs6EOWOFdmc5kGw0CDYdnfX8rodohgUbyb_P0MeQwSFE9_pchtjk0tatb7cdDCwjUMrbExGoSPSRZiDHrllB6d6pW32qiZ2wBzjcpAlPteWKohGX7iAnbQPJ9tCp4Rj_9buflbuX_WVjjg=&uniplatform=NZKPT&language=CHS]. 

[47]: We are sorry that we did not find the DOI of this reference. But we have provided the link here: https://kns.cnki.net/kcms2/article/abstract?v=_W1AupcyYgb813IaIi9LDlH_JKnWS4CNXgPCigdAjfQaBHFPMQweE4viS_iMn8P29SyXRAGdPzRzeniRyJ5UCt_-A5vxyE-puwj32dbPYZ3vZTBgpq1VROorCmb6kqCVYGJo7VY6-k1Fj6hVXzUVu-Sc7RgXkmLJeYd0n3MfyYZMk0zQgd0n4eyU6sgBlWAg&uniplatform=NZKPT&language=CHS. 

[48]: We are sorry that we did not find the DOI of this reference. But we have provided the link here: https://kns.cnki.net/kcms2/article/abstract?v=_W1AupcyYgbBBbNdvDZM7u4fc2TSHUn37i6Sjk7ds3RkcPAyXpYfvCNhWUKrmY-Ode0-UVHDpSiF2Zf_GqjUehT4nJUuOkUmQYJCBe8R0-WDwgoFQRnifw7d_eW2L5vnXW71aT5F2sR2AEwkX-1qX0C39MzQBmXRNLe8hhaB7XA5kDFMxkkCYKAQgdl-e7AL&uniplatform=NZKPT&language=CHS. 

[49]: We added DOI. From[Nikolaieva N, Brindza J, Garkava K, Ostrovský R. Pollen features of hazelnut (Corylus avellana L.) from different habitats. Modern Phytomorphology. 2014;6:53-8] to [Nikolaieva N, Brindza J, Garkava K, Ostrovský R. Pollen features of hazelnut (Corylus avellana L.) from different habitats. Modern Phytomorphology. 2014;6:53-8. doi: :10.5281/zenodo.160440].

[50]: We added DOI. From[Li JJ, Zhang R, Q., Ma QH, Wang GX. SEM observation on the pollen morphology in Corylus. Journal of Chinese Electron Microscopy Society. 2017;36(04):404-13] to [Li JJ, Zhang R, Q., Ma QH, Wang GX. SEM observation on the pollen morphology in Corylus. Journal of Chinese Electron Microscopy Society. 2017;36(04):404-13. doi: 10.3969/j.issn.1000-6281.2017.04.014].

[56]: We added DOI. From[Matthäus B, Özcan MM. The comparison of properties of the oil and kernels of various hazelnuts from Germany and Turkey. European Journal of Lipid Science and Technology. 2012;114:801-6] to [Matthäus B, Özcan MM. The comparison of properties of the oil and kernels of various hazelnuts from Germany and Turkey. European Journal of Lipid Science and Technology. 2012;114:801-6. doi: https://doi.org/10.1002/ejlt.201100299].

Reviewer #1:

1. Reference Formatting: As you pointed out, some of the newly added references do not include the journal names, page numbers, or volumes. The authors should standardize their references in line with the required citation style of PLOS ONE. Ensuring complete and consistent referencing, including missing elements (e.g., journal name, pages, volume), is critical for a high-quality academic manuscript.

The author’s answer:

The following are the details of the revision of the reference (The following numbers are reference numbers, with changes highlighted in red in the revised manuscript):

[2]: We added the number of pages. From[Brown JA, Beatty GE, Montgomery WI, Provan J. Broad-scale genetic homogeneity in natural populations of common hazel (Corylus avellana) in Ireland. Tree Genet Genomes. 2016;12(6). doi: ARTN 12210.1007/s11295-016-1079-7. PubMed PMID: WOS:000397238800022.] to [Brown JA, Beatty GE, Montgomery WI, Provan J. Broad-scale genetic homogeneity in natural populations of common hazel (Corylus avellana) in Ireland. Tree Genet Genomes. 2016;12(6):122. doi: 10.1007/s11295-016-1079-7. PubMed PMID: WOS:000397238800022].

[5]: We made the changes according to the requirements of the journal reference format. From[Zong JW. Study on Population Genetic Diversity of three Chinese Corylus Species and the Phylogentic Relationship of Corylus Species [PhD]: Chinese Academy of Forestry; 2017] to [Zong JW. Study on Population Genetic Diversity of three Chinese Corylus Species and the Phylogentic Relationship of Corylus Species. PhD. Thesis, Chinese Academy of Forestry; 2017. Available from: https://kns.cnki.net/kcms2/article/abstract?v=_W1AupcyYgZJYRJtAWj7Rty7HrHEdpbxgn9Qg1m4Msdu9fvGGxf9aUbhf2a3hQcSgkSxk4JKLgzHWheDOdXcswqGUsQHPO-ao1virqxCWge0oY-54IJw4MYaJ3DkWGTFAmDqHWWm1FhHUVd6nhN8dLBylgj1YZ4t5VzU51_e_NBlxQAHapTAcZPaD_WUuDDDfWXnsjfLcyI=&uniplatform=NZKPT&language=CHS].

[12]: We added DOI. From[Luo F, Fei XQ, Tang FB, Li XL. Simultaneous determination of paclitaxel in hazelnut by HPLC-MS/MS. Forest Research. 2011;24:779-83] to [Luo F, Fei XQ, Tang FB, Li XL. Simultaneous determination of paclitaxel in hazelnut by HPLC-MS/MS. Forest Research. 2011;24:779-83. doi: https://doi.org/10.1016/j.heliyon.2023.e13675].

[17]: We made the changes in accordance with the format requirements of the reference "Book chapters" in the journal. From[Vahdati K, Sheikhi A, Arab MM, Sarikhani S, Habibi A, Ataee H. Cultivars and Genetic Improvement. In: Mir MM, Rehman MU, Iqbal U, Mir SA, editors. Temperate Nuts. Singapore: Springer Nature Singapore; 2023. p. 79-111] to [Vahdati K, Sheikhi A, Arab MM, Sarikhani S, Habibi A, Ataee H. Cultivars and Genetic Improvement. In: Mir MM, Rehman MU, Iqbal U, Mir SA, editors. Temperate Nuts. Singapore: Springer Nature Singapore; 2023. pp. 79-111].

[18]: We have added journal names, volumes, pages and DOI. From[Vahdati K, Khorami SS, editors. The past, present and future of walnut genetic improvement and propagation2021: International Society for Horticultural Science (ISHS), Leuven, Belgium] to [Vahdati K, Khorami SS, editors. The past, present and future of walnut genetic improvement and propagation. Acta Hortic. 1318: 251-258. doi: https://doi.org/10.17660/ActaHortic.2021.1318.37].

[19]: We made the changes in accordance with the format requirements of the reference "Book chapters" in the journal. From[Vahdati K, Arab M, Sarikhani S, Sadat-Hosseini M, Leslie C, Brown P. Advances in Persian Walnut (Juglans regia L.) Breeding Strategies. 2019. p. 401-72] to [Vahdati K, Arab M, Sarikhani S, Sadat-Hosseini M, Leslie C, Brown P. In: Advances in Persian Walnut (Juglans regia L.) Breeding Strategies. Al-Khayri, J., Jain, S., Johnson, D. (eds) Advances in Plant Breeding Strategies: Nut and Beverage Crops. Springer, Cham; 2019. pp. 401-72].

[25]: We added pages. From[Zhao T, Ma W, Yang Z, Liang L, Chen X, Wang G, et al. A chromosome-level reference genome of the hazelnut, Corylus heterophylla Fisch. Gigascience. 2021;10(4). Epub 2021/04/20. doi: 10.1093/gigascience/giab027. PubMed PMID: 33871007; PubMed Central PMCID: PMCPMC8054262] to [Zhao T, Ma W, Yang Z, Liang L, Chen X, Wang G, et al. A chromosome-level reference genome of the hazelnut, Corylus heterophylla Fisch. Gigascience. 2021;10(4): 1-9 Epub 2021/04/20. doi: 10.1093/gigascience/giab027. PubMed PMID: 33871007; PubMed Central PMCID: PMCPMC8054262].

[29]: We have added journal names, volumes, pages and DOI. From[Vahdati K, Mohseniazar M, editors. Early bearing genotypes of walnut: a suitable material for breeding and high density orchards2016: International Society for Horticultural Science (ISHS), Leuven, Belgium] to [Vahdati K, Mohseniazar M, editors. Early bearing genotypes of walnut: a suitable material for breeding and high density orchards. Acta Hortic. 1139: 101-106. doi: https://doi.org/10.17660/ActaHortic.2016.1139.18].

[30]: We have added journal names, volumes, pages and DOI. From[Vahdati K, Karimi R, Ershadi A, editors. GENETIC STRUCTURE OF SOME WILD WALNUT POPULATIONS IN IRAN2015: International Society for Horticultural Science (ISHS), Leuven, Belgium] to [Vahdati K, Karimi R, Ershadi A, editors. GENETIC STRUCTURE OF SOME WILD WALNUT POPULATIONS IN IRAN. Acta Hortic. 1074: 125-128. doi: https://doi.org/10.17660/ActaHortic.2015.1074.18]

[34]: We are sorry that we did not find the DOI of this reference. But we have provided the link here: https://kns.cnki.net/kcms2/article/abstract?v=_W1AupcyYgZACCJ7g5T4YcpuIJNx0YaDr-dpt2j_vUqw28MX81RkRe2ARt84fqqCapqeVWZKnPHbRwU22_l7vVtPkwdAXJy2TSWpUQxHh-JjQEWzHfpG2GaU-Ul5OAJ0suup9_2ie4FJ5iMF8yqTsKKCPnF8gff7uNQlT7gemRg0L4_nuRm1Qhnwt8v8FP_P&uniplatform=NZKPT&language=CHS.

[35]: We added DOI. From[Long ZY, Lu CH. Resourse distribution and development research progress of Corylus in Heilongjiang province. Forest By-Product and Speciality in China. 2005;(04):41-2] to [Long ZY, Lu CH. Resourse distr

---

## [Decision Letter · Decision Letter 2]

4 Nov 2024

Characteristics of wild hazelnut populations in northeast China and selection of superior provenances

PONE-D-24-23349R2

Dear Dr. Guan,

We’re pleased to inform you that your manuscript has been judged scientifically suitable for publication and will be formally accepted for publication once it meets all outstanding technical requirements.

Kind regards,

Ali Akbar Ghasemi-Soloklui, Ph.D

Academic Editor

PLOS ONE

 <!--</p

Reviewers' comments:

Reviewer's Responses to Questions

**Comments to the Author**

1. If the authors have adequately addressed your comments raised in a previous round of review and you feel that this manuscript is now acceptable for publication, you may indicate that here to bypass the “Comments to the Author” section, enter your conflict of interest statement in the “Confidential to Editor” section, and submit your "Accept" recommendation.

Reviewer #1: All comments have been addressed

2. Is the manuscript technically sound, and do the data support the conclusions?

Reviewer #1: Yes

3. Has the statistical analysis been performed appropriately and rigorously? 

Reviewer #1: Yes

4. Have the authors made all data underlying the findings in their manuscript fully available?

Reviewer #1: Yes

5. Is the manuscript presented in an intelligible fashion and written in standard English?

Reviewer #1: Yes

6. Review Comments to the Author

Reviewer #1: The authors revised the paper according to the reviewers' comments thoroughly and the paper can be considered as an accepted paper.

7. PLOS authors have the option to publish the peer review history of their article (what does this mean?). If published, this will include your full peer review and any attached files.

Reviewer #1: No

---

## [Editor Report · Acceptance letter]

20 Nov 2024

PONE-D-24-23349R2 

PLOS ONE

Dear Dr. Guan, 

I'm pleased to inform you that your manuscript has been deemed suitable for publication in PLOS ONE. Congratulations! Your manuscript is now being handed over to our production team.

Kind regards, 

on behalf of

Dr. Ali Akbar Ghasemi-Soloklui 

Academic Editor

PLOS ONE